

# An Evaluation of three methods for measuring black carbon at Alert, Canada

Sangeeta Sharma[1], W. Richard Leaitch[1], Lin Huang[1], Daniel Veber[1], Felicia Kolonjari[1], and Wendy Zhang[1]

[1]Climate Chemistry Measurements and Research, Climate Research Division, Atmospheric Science and Technology Directorate, Environment and Climate Change Canada, 4905 Dufferin Street, Toronto, ON M3H 5T4, Canada

Sarah J. Hanna[2] and Allan K. Bertram[2]

[2]Department of Chemistry, University of British Columbia, Vancouver BC, Canada

John A. Ogren[3, 4]

[3]Cooperative Institute for Research in Environmental Sciences, University of Colorado, Boulder, CO 80309 USA
and [4]NOAA/ESRL Global Monitoring Division, 325 Broadway R/GMD, Boulder, CO 80305 USA

*Correspondence to:* Sangeeta Sharma (sangeeta.sharma@canada.ca)

Abstract

Absorption of sunlight by black carbon (BC) warms the atmosphere, which may be important for Arctic climate. The measurement of BC is complicated by the lack of a simple definition of BC and the absence of techniques that are uniquely sensitive to BC (e.g. Petzold et

al., 2013). At the Global Atmosphere Watch baseline observatory at Alert, Nunavut (82.5$^{o}$N), BC mass is estimated in three ways, none of which fully represent BC: conversion of light absorption measured with an Aethalometer to give equivalent black carbon (EBC), thermal desorption of elemental carbon from weekly-integrated filter samples to give elemental carbon (EC), and measurement of incandescence from the refractory black carbon (rBC) component of

individual particles using a Single Particle Soot Photometer (SP2). Based on measurements between March 2011 and December 2013, EBC and EC are 2.7 and 3.1 times higher than rBC respectively. The EBC and EC measurements are influenced by factors other than just BC, and higher estimates of BC are expected from these techniques. Some bias in the rBC measurement may result from calibration uncertainties that is difficult to estimate here. Considering a number

of factors, our best estimate of BC mass at Alert, which may be useful for evaluation of chemical transport models, is an average of the rBC and EC measurements with a range bounded by the rBC and EC combined with the respective measurement uncertainties. Winter, spring, summer, and fall averaged (± atmospheric variability) estimates of BC mass at Alert for this study period are 49±28 ng m$^{-3}$, 30±26 ng m$^{-3}$, 22±13 ng m$^{-3}$, and 29±9 ng m$^{-3}$, respectively. Average coating

thicknesses estimated from the SP2 are 25% to 40% of the 160-180 nm diameter rBC core sizes. For particles of approximately 200-400 nm optical diameter, the fraction containing rBC cores is estimated to be between 10% and 16%, but the possibility of smaller undetectable rBC cores in





some of the particles cannot be excluded. Mass absorption coefficients (MAC)±uncertainty at 550 nm wavelength, calculated from light absorption measurements divided by the best estimates of BC mass concentrations, are $8\pm4$, $8\pm4$, $5\pm2.5$ and $9\pm4.5$ m$^2$ g$^{-1}$, for winter, spring, summer and fall respectively. Adjusted to better estimate absorption by BC only, the winter and spring values

of MAC are $7.6\pm3.8$ and $7.7\pm3.8$ m$^2$g$^{-1}$. There is evidence that the MAC values increase with coating thickness.

## 1. Introduction

Black carbon (BC) is a component of the atmospheric aerosol that strongly absorbs

shortwave radiation. A comprehensive review suggests the impact of BC on direct radiative forcing of the atmosphere is 0.71 W m$^{-2}$, with an uncertainty range of +0.08 to +1.27 W m$^{-2}$ (Bond et al., 2013). BC is a short-lived climate forcer (SLCF) due to its relatively short atmospheric lifetime of a few days to a few weeks. It has been suggested that mitigation of BC emissions may reduce warming of the Arctic atmosphere in the short term (UNEP/WMO, 2011;

AMAP 2015; Sand et al., 2015). BC results from incomplete combustion of carbonaceous fuels, and the definition of BC is complex (Andreae and Gelencsér, 2006; Bond et al., 2013; Petzold et al., 2013). Most of the BC in Arctic aerosol particles is transported to the Arctic from lower latitude sources during winter and spring (e.g. Barrie, 1986; Sharma et al., 2004; Stone et al., 2015). The Arctic atmosphere is relatively stable, resulting in pollution transport into the Arctic

often occurring in layers. BC in particles will warm the atmospheric layer in which they reside, while the reflective components of the particles (e.g. sulfate, non-absorbing components of organics) cool the atmosphere and surface below the layer. The degree of heating of the layer and cooling of the surface below depends in part on the albedo of the surface below: surfaces with relatively high albedos (snow, ice and clouds) are cooled less and could enhance warming by the

absorbing layer. Aerosol particles containing BC in layers well above the surface will tend to increase the stability of the Arctic atmosphere (e.g. Brock et al., 2011), whereas those transported near the surface may warm the air over highly reflective surfaces and even the less reflective surfaces that are found in the Arctic during summer (e.g. Iziomon et al., 2006). Deposition of BC can lower the albedo of snow and ice-covered areas of the Arctic, making

another contribution to Arctic warming (Clarke and Noone, 1985; McConnell et al., 2007; Hegg et al., 2009; Keegan et al., 2014; Dumont et al., 2014). Outside of the Arctic, BC can alter



latitudinal temperature gradients, which may be more important for Arctic warming than absorption within the Arctic (Sand et al., 2013).

Measurements of BC in the Arctic are relatively scarce; our long-term knowledge of BC has been based on light-absorption measurements of particles that are converted to EBC mass
concentrations using a mass absorption coefficient (MAC) (e.g. Sharma et al., 2004; 2006; Eleftheriadis et al., 2009; Massling et al., 2015). BC is insoluble in water and organic solvents, and it is refractory to over $3600^{o}$C (Schwarz et al., 2006; Petzold et al., 2013). Freshly emitted BC particles often exhibit complex morphologies that change as the BC becomes internally mixed with other aerosol components; this aging process can alter the absorption properties (e.g.
Petzold et al., 2013).  Particle absorption is also affected by dust and by brown carbon (BrC), where the latter arises from anthropogenic sources (e.g. Petzold et al., 2009; 2011) and biomass burning (e.g. Hoffer et al., 2006; Andreae and Gelencsér, 2006).  In addition to the presence of other absorbing components of the atmospheric aerosol, light absorption measurements used to estimate BC are complicated by absorption enhancement by the filtering media and the
uncertainty in the MAC value. Thus, more direct techniques to measure BC are necessary.

Estimates of BC mass concentrations are made at the Dr. Neil Trivett Global Atmospheric Watch Observatory at Alert, Nunavut, Canada ($82.5^{o}$N and $62.5^{o}$W; 185 m-MSL) using three approaches. Since May 1989, light absorption by particles has been measured with an Aethalometer (Hopper et al., 1994; Sharma et al., 2004; 2006). The light absorption is converted
to BC using a MAC value, which is an indirect method for estimating BC referred to as Equivalent Black Carbon (EBC) (Petzold et al., 2013). Weekly-averaged collections of particles on quartz filters were initiated in 2005. The filters are subsequently analyzed for elemental carbon (EC) and organic carbon (OC) using an in-house thermal technique known as EnCan-total-900 (Huang et al., 2006; Chan et al., 2010). In 2011, a Droplet Measurements
Technologies, Inc. Single Particle Soot Photometer (SP2) (Stephens et al., 2003; Schwarz et al., 2006), was installed at Alert enabling real-time measurements of refractory black carbon (rBC) based on the incandescence of individual particles heated to $3600^{o}$C. Also light absorption measurements in three wavelengths were made with a Particle Soot Absorption Photometer (PSAP; Radiance Research, Inc.). An instrument based on acoustic detection was also deployed





at Alert to estimate BC, but the necessary sensitivity in the relatively low concentration environment could not be achieved and it is not included in the discussion here.

The objective of this paper is to improve the characterization of BC and MAC values in the high Arctic. Presented here are comparisons of EBC, EC, and rBC observations made at the Alert observatory for the period from Mar 2011 to Dec 2013. In addition, number and mass size distributions, as well as coating thicknesses of the rBC particles, are discussed. These results are the first reported measurements of rBC over an extended period at any high Arctic location as well as the first seasonal comparison of rBC to EBC and EC. In section 2, the measurement techniques and analysis methods are discussed. Section 3 presents time series and seasonal variations of masses of rBC, EBC, and EC as well as rBC number size distributions and coating thicknesses. The discussion in section 4 addresses some questions raised by the results of this work, such as: why are the responses of instruments to aerosol black carbon different at Alert; how does the MAC value vary seasonally; and how do these results compare with other related studies? Section 5 summarizes the findings and identifies the conclusions.

## 2. Methods

### 2.1. Sampling

The Alert Observatory is a global station within the World Meteorological Organization Global Atmospheric Watch program (Fig. 1), and the sampling protocols follow the GAW recommendations (WMO, 2016). The aerosol intake is at a height of 10 m above the ground, and the particles are pulled down a 20 cm diameter vertical tube at a flow rate of 1000 L min$^{-1}$. The particles in the centre of the air stream pass through a 2.5 cm diameter stainless steel tube which is heated, as needed, to maintain a relative humidity (RH) of no more than 40% at a flow rate of 120 L min$^{-1}$. The flow is further split into four ¾-inch (1.9 cm) diameter stainless steel tubes. The flows for all measurements discussed here are drawn from this common inlet. Table 1 summarizes list of all instruments.

### 2.2. Optical measurements

The optical measurements of light absorption by the aerosol particles require corrections due to scattering and absorption effects from the filter media and particle loading of the filters. Further, conversion of the light absorption estimate to a BC mass concentration requires





knowledge of the value of the MAC. The MAC will vary depending on the morphology of the BC component of the particle as well as the nature of the other components in the particle (Bond et al., 2013). As a consequence of the indirect nature of this BC estimate, it is referred to as equivalent black carbon or EBC. Here, EBC is derived from the Aethalometer only. Light

absorption measurements are taken by the Particle Soot Absorption Photometer (PSAP).

### 2.2.1. Aethalometer measurements of EBC

    EBC mass is estimated from particle light absorption measured with a Magee Scientific

AE-31 Aethalometer (Hansen et al., 1984). The Aethalometer measures the real-time attenuation of light transmitted through particles accumulating on a quartz fiber filter (reinforced quartz fiber tape) at 7 wavelengths. Measurements are accumulated at 5 min intervals. A vacuum pump draws air through the instrument so that the particles continuously accumulate on the filter while being illuminated. The effective operational wavelengths of Aethalometer are 370, 470, 520,

590, 660, 880 and 950 nm. The EBC measurements used here are based on the 880 nm wavelength to minimize potential interference from other absorbing components (e.g. brown carbon). The use of the 880 nm wavelength also maintains continuity with historical EBC data measured by single 880 nm wavelength Aethalometer used at Alert from 1989 to 2009 (e.g. Sharma et al., 2004; 2006; 2013).

The intensity of light transmitted through the filter is measured by two photo-diodes: one through the sample spot ($I_s$) and the other through a blank (unsampled) portion of the filter called the reference spot ($I_r$). The filter attenuation is defined as

$$ATN = -ln\left(\frac{I_s}{I_r}\right) \qquad (1)$$

The change in attenuation is obtained as a function of time and relates to the EBC concentration as follows:

$$\sigma_{atn} = -\frac{A}{Q}\frac{\left(ATN(t_2) - ATN(t_1)\right)}{(t_2 - t_1)} \qquad (2)$$

$$EBC = -\frac{\sigma_{atn}}{\alpha_{ap}} \qquad (3)$$





Where ATN(t) is the filter attenuation at times $t_1$ and $t_2$ (in seconds); Q (m³ s⁻¹) is the sample flow rate through the filter; A (m²) is the area of the exposed spot on the filter; $\sigma_{atn}$ is the attenuation coefficient, and $\alpha_{ap}$ is the specific attenuation coefficient (m² g⁻¹). The manufacturer's recommended value for $\alpha_{ap}$ is 14625/λ, which is based upon calibrations during instrument

development and theoretical calculations. It has a value of 16.6 m² g⁻¹ at λ=880 nm. This accounts for absorption by BC and additional light attenuation assumed to be caused by multiple scattering within the filter media. There are no other scattering or loading corrections applied to Aethalometer data because a comparison of unmodified EBC mass to rBC mass values are also determined in this paper. The detection limit of the Aethalometer is dependent on the stability of

the optics. Changes in the light intensity correspond to a noise level of 2 ng m⁻³. Taking twice the standard deviation in the noise, we report a detection limit of 4 ng m⁻³ for a 1-hour integration time estimated from bench zeros ran at the ECCC laboratory for 3 days of particle free air and a value of 2.2 ngm⁻³ for 1- hour integration time as estimated from bench zeros at Alert as also used in the Backman et al. (2017) analysis.

### 2.2.2. Particle Soot Absorption Photometer (PSAP) for light absorption

The PSAP utilizes a similar principle in its operation to the Aethalometer (Bond et al., 1999), measuring aerosol absorption at three wavelengths (467, 530, and 660 nm). An algorithm

for correcting the attenuation coefficient measured by the PSAP to the aerosol absorption coefficient, $\sigma_{ap}$, was derived by Bond et al., 1999) and further refined by Ogren (2010):

$$\sigma_{ap} = 0.85 \frac{f(\tau)\,\sigma_{atn}}{K_2} - \frac{K_1 \sigma_{sp}}{K_2} \quad (4)$$

where $\sigma_{sp}$ is the aerosol light scattering coefficient adjusted to the wavelength of the absorption measurement. The transmittance correction term is defined as $f(\tau)=(1.0796\tau+0.71)^{-1}$, where

$\tau = (I_s(t)/I_r(t))/(I_s(t=0)/I_r(t=0))$ is the filter transmittance at time $t$ relative to the unsampled filter at time $t$=0. The constants in equation 4 were derived by Bond et al. (1999) as $K_1$=0.02±0.02 and $K_2$=1.22±0.20.

The PSAP absorption measurements at 530 nm have been converted to 550 nm absorption for this study. The exposed spot on which the sample is collected is 0.5 cm in

diameter for the PSAP, compared to 1.1 cm for the Aethalometer. The detection limit for the





PSAP, defined as the noise level for a 60 s sampling interval, was determined to be 0.2 Mm$^{-1}$ for a one-minute integration time (twice the standard deviation). The hourly detection limit is estimated to be 0.08 Mm$^{-1}$. This was determined at the site with regular two-hour, weekly zero checks by passing particle-free air through all instruments including the PSAP. Adjustments for

changes in the flow and spot area have been applied to the data.

## 2.2 Single Particle Soot Photometer (SP2) for rBC

Measurements of refractory black carbon (rBC) in single particles were obtained using a

Single Particle Soot Photometer (SP2, Droplet Measurement Technologies Inc., Boulder, CO). The rBC data were collected from three different models of SP2 instruments during three different time-periods as given in Table 1. No discontinuities are evident in the data before and after the instrument changes.

The detailed operating principles of an SP2 have been described previously (Stephens et

al., 2003; Baumgardner et al., 2004; Schwarz et al., 2006); therefore, only a brief overview is given here. Particles are directed into the SP2 where they intersect a continuous, high intensity ($10^6$ MW/cm$^2$), intra-cavity Nd:YAG laser beam, operating at 1064 nm. Particles intersecting the laser beam can both scatter and absorb light. Particles with a component that absorbs at 1064 nm are heated and begin to incandesce as they absorb the incident radiation. The rBC mass

concentration of a single particle is determined from the strength of the incandescence signal based on calibration using Aquadag particles of known size. For the calibration, a suspension of Aquadag in water is atomized and dried, and the dried Aquadag particles are size selected with a differential mobility analyzer (Schwarz et al., 2010). On-site calibrations were conducted for all instruments. The mobility diameter of the calibration particles was converted to rBC mass using

the parameterization developed by Gysel and colleagues (Gysel et al., 2011). Recent studies have shown that the SP2 is more sensitive to Aquadag than it is to other types of black carbon and calibrations with Aquadag can underestimate the ambient rBC mass concentrations (Laborde et al., 2012; Moteki and Kondo, 2010). To account for this, the Aquadag calibration curves were scaled by a factor of $0.70 \pm 0.05$ based on the work of Laborde et al. (2012). They determined

this scaling factor based on the relative sensitivity of the SP2 to Aquadag compared with rBC from denuded ambient particles from diesel and wood smoke. An example of the calibration curves for SP2 #58 (both low-gain and high-gain channels) are shown in Fig. 2 for soot particle



masses up to 41 fg. The figure includes combined data from four calibrations carried out on 6 Nov 2012, 30 Aug 2014, 11 Apr 2015 and 1 Dec 2015.

For comparison to the 1 μm sample size-cuts used in front of the filters for thermal analysis (EC) and the PSAP measurements, the rBC mass over the range of 80-1000 nm was
estimated by fitting a single lognormal distribution to each measured particle mass distribution. As an example, Fig. 3a and 3b show the seasonally-averaged rBC mass and number distributions and the fitted distributions. In this example, the measurements, indicated by the circles, are limited to 333 nm VED due to the averaging involving the 4-channel SP2 (#17). Overall, the fitted distributions are reasonable approximations.  The discontinuities are associated with the
mixing of the different instruments and years in deriving these averaged curves. The  number distributions are estimated from the fits to the mass distribution and will be discussed in section 3.2.1.

## 2.3 Thermal method (EnCan-total-900) for EC

Weekly-integrated samples of particles collected on quartz filters with a 1μm upper size cut were analyzed for elemental carbon (EC) and organic carbon (OC) using an in-house thermal technique referred to as EnCan-total-900. This technique was originally developed for carbon isotope analysis of OC and EC (Huang et al., 2006). This method differs from the thermal-optical
methods used in the Interagency Monitoring of Protected Visual Environments (IMPROVE) network (e.g. Chow et al., 2001) and by the National Institute for Occupational Safety and Health (NIOSH, 1996; 1999), as it does not incorporate laser reflectance or transmittance but only temperature and retention time used to determine OC and EC. The EnCan-total-900 technique involves three temperature dependent steps. The first two steps occur under a pure
helium condition at 550 °C for the detection of OC and at 870 °C for the detection of pyrolysis OC (POC) and carbonate carbon (CC). EC is detected at 900°C under helium and 10% oxygen. Compared to the IMPROVE and NIOSH methods, the retention times at each step are much longer: 600 s, 600 s and 420 s at 550 °C, 870 °C and 900 °C, respectively.  By introducing the 870°C pure helium phase, the POC and CC are released such that the effect of OC charring on
EC is minimized. An example thermograph from analysis of a NIST standard (SRM8785-urban dust) is shown in Fig. 4. Repeated measurements of SRM8785 over 6 years indicate an uncertainty in the EC measurements of <10% for this urban dust aerosol. EC determined by



thermal and thermal-optical methods is dependent on the methodology to some degree. An inter-laboratory comparison among different methods used in long-term atmospheric observation networks showed the relative standard deviation of the mean value of EC measurements in an inter-comparison effort by the three protocols, i.e., the IMPROVE, EUSAAR and EnCan-total-

900 to be 25 % (Karanasiou et al., 2015). Also, the EnCan-total-900 method has been verified by comparing the mass fractions of OC, EC, POC, and CC with the corresponding weighed amounts. The measurements of isotopic compositions ($^{13}C/^{12}C$ & $^{14}C/^{12}C$) indicate quantitative separation of OC and EC (Huang et al., 2015).

**2.4 Uncertainties in the measurement techniques**

**Aethalometer**

The relative uncertainty of the measured light attenuation coefficient is defined by Backman et al. (2017):

$$\frac{\delta\sigma_{atn}}{\sigma_{atn}} = \sqrt{\left(\frac{\delta\Delta ATN^2}{\Delta ATN^2}\right) + \frac{\delta A^2}{A^2} + \frac{\delta Q^2}{Q^2}} \qquad (5)$$

Backman et al. (2017) estimate that for $\Delta ATN \geq 2$ a relative uncertainty at 880 nm at Alert is 2.5% (for noise only) in a 24-hour time period. The uncertainty in flow rate is 1.5% as reported by the manufacturer of the flow controller and 2% for the spot size can be achieved by digital

image analysis. Backman et al. (2017) estimated 36% relative uncertainty of the instrument including the drift. There are uncertainties in $\sigma_{atn}$ for the Aethalometer due to particle loading and scattering that cannot be determined for this study. A constant $\alpha_{ap}$ value of 16.6 m$^2$ g$^{-1}$ is used to estimate black carbon mass from the Aethalometer measurements. It has been shown that there can be large uncertainties in the $\alpha_{ap}$ value (e.g. Louisse et al., 1993; Sharma et al., 2002).

**PSAP**

The main sources of uncertainty in the light absorption measurement from the PSAP are the measurement of the instrumental noise $\Delta\sigma_{ap,noise}$ sample spot size, the flow calibration $\Delta\sigma_{ap,flow,spotsize}$, and the uncertainty $\Delta\sigma_{ap,cal}$ of the calibration constants $K_1$ and $K_2$ in the Bond

et al. (1999) correction (Equation 4). The combined uncertainty for the PSAP measurements is





$$\Delta\sigma_{ap}{}^2 = \left(\Delta\sigma_{ap,cal}{}^2 + \Delta\sigma_{ap,flow,spotsize}{}^2 + \Delta\sigma_{ap,noise}{}^2\right) \tag{6}$$

The standard deviation in the 1-min and 1-hour absorption data for particle free air at Alert are

0.14 Mm$^{-1}$ and 0.005 Mm$^{-1}$ at 550 nm wavelength. The combined uncertainty, $\Delta\sigma_{ap,flow,spotsize}$,

depends on the uncertainty in flowmeter calibration (1.5%) and measurement of spot size (2%).

Sherman et al. (2015) showed that the uncertainty in absorption depends on the uncertainty in the

K1 and K2 values in the Bond et al. (1999) correction. Equation 7 is rewritten in Eq. 6 in terms

of the single scattering albedo, $\omega_0$ ($\sigma_{sp/( \sigma_{sp} + \sigma_{ap})}$).

$$\sigma_{ap} = \frac{\sigma_{ap,meas}}{a*K_1+K_1} \tag{7}$$

where $a = \omega_0/(1- \omega_0)$ and $\sigma_{ap,meas} = \sigma_{atn}*f(\tau)$ as defined above. The uncertainty in $\sigma_{ap}$ from

calibration constants is given by:

$$\Delta\sigma_{ap,cal} = \left(\left(\frac{\partial\sigma_{ap}}{\partial K_1}*\Delta K_1\right)^2 + \left(\frac{\partial\sigma_{ap}}{\partial K_2}*\Delta K_2\right)^2\right)\right)^{1/2} \tag{8}$$

Eq. 8 can be rewritten as follows:

$$\Delta\sigma_{ap,cal} = \frac{\sigma_{ap,meas}}{\left(0.02*\frac{\omega_0}{1-\omega_0}+1.44\right)^2} * \left(\left(0.02*\frac{\omega_0}{1-\omega_0}\right)^2 + (0.24)^2\right)^{\frac{1}{2}} \tag{9}$$

At Alert, $\omega_0$ calculated from the measurements range less than 1 and 0.95. For a $\omega_0$ of 0.95 and

light absorption values ($\sigma_{ap,meas}$) of 0.5 Mm$^{-1}$ and 1 Mm$^{-1}$ (as typical for Alert), Eq. 9 defines an

uncertainty in the absorption coefficient, which has been calculated to be 0.099 Mm$^{-1}$ and 0.11

Mm$^{-1}$ (i.e. 10-20%) respectively.

Weekly zeroes are performed on the PSAP at Alert by flushing particle-free air through the

instrument. During this process, the uncertainty in the measurement due to instrumental noise

was determined.





**SP2**

There are a number of uncertainties associated with estimating the rBC mass from the calibrations that were performed for sizes between 80 nm and 225 nm. First, the mass calibration is extrapolated from 225 nm to 1000 nm, but the linear correlations between rBC mass and peak

height are relatively strong, as shown in Fig. 2, suggesting that this is not a large source of error. Second, there are uncertainties due to the density assumption used to infer individual rBC particle mass from mobility-based size selection during a calibration as well as the scaling factor used to correct for the difference in sensitivity of the SP2 to Aquadag and ambient rBC (the Aquadag correction factor). Third, there is uncertainty associated with fitting the mass

distribution and using this fit to estimate how much rBC mass lies outside the instrument detection range (80-500 nm). Considering the contributions arising from the extrapolation of the calibration curve, the Aquadag correction factor, and the fit of the mass distribution, the estimated uncertainty in the 1-hour mass concentration for SP2#58, SP2#44, and SP#17 is 19%, 18%, and 23%, respectively. Unaccounted for is the uncertainty in the density of the Aquadag

particles used for calibrations.  Assuming the uncertainty in the 0.7 g cm$^{-3}$ density is $\pm0.2$ g cm$^{-3}$, or 29%, the overall uncertainty in the rBC mass is 35%. In addition, if a large number of rBC cores are smaller than 80 nm, rBC mass will be under-estimated.

**Thermal technique**

20       The EC mass concentration from the filter analyses is calculated as follows:

$$C = X * \frac{A}{V} \qquad (8)$$

Where X is the area concentration ($\mu$gC/cm$^2$) of the filter punch analyzed by the OC-EC analyser (Sunset Lab Inc. www.sunlab.com); A is the sampling area (cm$^2$) of a quartz filter with a diameter of 47mm (Pall Corporation, Analyslide$^{TM}$ Petri Dishes); V is the total integrated air

volume (m$^3$) sampled through the filter; C is the concentration of the integrated air sample on the filter ($\mu$gC/cm$^3$). The relative uncertainty is estimated by:

$$\frac{\partial C}{C} = \sqrt{\left(\frac{\partial X}{Xi}\right)^2 + \left(\frac{\partial B}{Xi}\right)^2 + \left(\frac{\partial A}{A}\right)^2 + \left(\frac{\partial V}{V}\right)^2 + \left(\frac{\partial X\_dup}{AX\_dup}\right)^2} \qquad (9)$$



Where $\left(\frac{\partial X}{Xi}\right)$ is the relative uncertainty of instrument measurement for carbon mass, based on the accuracy and precision determined by the calibrations over the period of 2010 to 2016, using a gravimetric approach on a sucrose standard; $\left(\frac{\partial B}{Xi}\right)$ is the relative uncertainty of EC contributed from field blank; $\left(\frac{\partial A}{A}\right)$ is the relative uncertainty of sampling area; $\left(\frac{\partial V}{V}\right)$ is the relative uncertainty of total sampled air volume; $\left(\frac{\partial X\_dup}{AX\_dup}\right)$ is the relative uncertainty of EC due to sampling inhomogeneity, based on duplicated analysis across the entire net-work. Among the five components, $\left(\frac{\partial X}{Xi}\right)$, $\left(\frac{\partial A}{A}\right)$, $\left(\frac{\partial V}{V}\right)$ and $\left(\frac{\partial X\_dup}{AX\_dup}\right)$ are the same for all the samples: 0.05, 0.07, 0.01, 0.1, respectively. While the term, $\left(\frac{\partial B}{Xi}\right)$, is individual sample dependent, the mean relative uncertainty of EC measurements at Alert over the period of March 2011 to December 2013 is approximately 28% and can be as high as 80% during summer due to very low EC concentrations.

## 3. Results
### 3.1 Time series and seasonal variations of masses of rBC, EBC and EC

Time series of the mass concentrations of rBC, EBC, and EC for March 2011 to December 2013 are shown in Fig. 5a-c, where EBC is derived from the Aethalometer at a wavelength of 880 nm. The rBC and EBC data are hourly averages, while the EC data are weekly integrated values. Over the study period, the mean rBC, EBC and EC are 17 ng m$^{-3}$, 38 ng m$^{-3}$, and 29 ng m$^{-3}$, respectively. Fig. 5d shows the rBC and EBC, after averaging to the EC weekly integrated times and subsequently monthly averages, with the monthly averaged EC. The monthly averaged rBC concentrations are lowest, EBC are highest and EC falls in between except for the summer (JJA) when EC is highest. The higher winter and spring values are the result of pollution transported to the Arctic from various anthropogenic sources at lower latitudes, a phenomenon often referred to as Arctic Haze (e.g. Barrie, 1986; Sharma et al., 2006, Quinn et al., 2007; Stone et al., 2015). Summertime concentrations are much lower than other seasons largely due to wet scavenging (e.g. Garrett et al., 2011; Croft et al., 2016). Previous characterizations of the pollution source regions influencing Alert indicate the potential source contribution function highest for Western Siberia followed by Europe and a very small influence at the surface from North America and Eastern Asia during winter and spring (Sharma et al.,





2004; 2006; 2013; Gong et al., 2010; Hirdman et al., 2010). More recently, global simulations suggest a broad influence of Eastern Asia at Alert that is strongest during spring and has a long transport time (Xu et al., 2017; Qi et al., 2017).

Seasonal averages of all measurements and their 25[th] and 75[th] percentiles are given in
Table 2 to show how all techniques respond to seasonal variation of the atmospheric changes in black carbon levels. The average EBC masses are significantly higher than the rBC masses ($p<0.01$) for all seasons by approximately a factor of two. Slopes, coefficients of determination and significance levels for linear regressions among EC, rBC, and EBC are given in Table 2. All results are significant at the 95% confidence level with the exception of winter EBC vs EC.

Table 3 gives mean±std dev and median values of EC, rBC and EBC for the entire study period to elucidate on differences in these techniques. Statistics are given for all data, data only above detection limit and only for pairwise data available i.e. data when both variables were available for comparison. The ratios of EBC and EC to rBC concentrations are approx. a factor of 3 higher for all data when only pairwise data points were considered.

Before discussing possible reasons for the differences among EBC, EC, and rBC, the rBC number size distributions and thicknesses of coatings associated with rBC particles as derived from the SP2 measurement are examined.

### *3.2 rBC Number Size Distributions and rBC Coating Thicknesses*

3.2.1 rBC Number Distributions

The rBC number distributions derived from the curves fitted to the rBC mass distributions are shown in Fig. 3b, using a density assumption for the ambient rBC of 1.8 g cm[-3] from Bond and Bergstrom (2006). Relative to the mass distributions, the uncertainty in the number distributions is greater below the lower limit of 80 nm than above the upper limit of 530 nm. Comparisons of
the seasonal number distributions indicate that both the mean concentrations and sizes of the rBC components of particles are larger during winter-spring than summer-fall. This pattern is consistent with increased wet scavenging of larger particles during summer-fall seasons (Garrett et al., 2011; Sharma et al., 2013).

3.2.2 rBC Coating Thicknesses



BC particles are often coated by other components (e.g. sulphate and organics, water) that can enhance the absorption by BC by increasing the light intercepted by the particle, sometimes referred to as a 'lensing effect' (e.g. Bohren and Huffman, 1983, Isaac et al., 1986; Cross et al., 2010; Shiraiwa et al., 2010). Shiraiwa et al. (2010) showed that even small coatings ($D_p/D_c = 1.2$,

where $D_p$ is the outer particle diameter and $D_c$ is the diameter of the core BC component, based on the core-shell concept) may result in an amplification of absorption of as much as 1.3, and the amplification for a $D_p/D_c$ of 2 is about 2. Thus, it is important to know the thickness of material coating the BC components, in addition to the index of refraction of the coating. To derive the coating thickness from the SP2 measurements, the scattering signal from incandescent particles

must first be reconstructed, for which the leading edge optimization (LEO) method of Gao et al. (2007) was used. Using Mie theory and assuming a core-shell model, the thickness of the coating present on the rBC core was calculated based on the measured scattering signal in conjunction with the measured rBC mass. The refractive index used for the core is 2.26– 1.26i (Moteki et al., 2010; Taylor et al., 2014) and the refractive index used for the coating material is 1.5–0.0i

(Metcalf et al., 2012; Schwarz et al., 2008), since the calibrations of the scattering signal were done with polystyrene latex spheres (PSLs). It should be noted that the assumption of concentric core-shell morphology is a simplification for rBC particles in the atmosphere (e.g., Moffet et al, 2016).

The SP2 simultaneously measures incandescence and light scattering by individual

particles with optical diameters in the range of approximately 200-400 nm, enabling coating thicknesses to be calculated for a limited but significant particle size range. Typically, much of the light extinction by fine particles occurs in this size range. The period when such analysis is possible depends on the availability of light scattering calibrations; for this work, the periods of analysis are limited to April 2012 and October-November 2013. The results of the coating

analyses, averaged over these two periods, are shown in Fig. 6 as a function of the rBC diameter. Fig. 6b shows the minimum coating thickness averaged for October-November, and Fig. 6c shows the maximum thickness for the same period. The maximum and minimum are shown because only two scattering calibrations were done for the SP2 in use at that time: one in November 2012 and one in December 2015. During the time between the two calibrations, there

was reduction in the light scattering signal by a factor of two from the 240 nm polystyrene latex (PSL) particle. Therefore, the calculations were done for each calibration under the assumption





that one yields a minimum coating thickness and the other a maximum coating thickness. The red dots in each panel indicate the fraction of rBC cores that could have a thickness estimate assigned. This fraction decreases with decreasing size as the ability to detect light scattered from a particle also decreases. Although the incandescence measurements can size rBC cores down to

approximately 80 nm, the elastic scattering optics in the SP2 can only detect bare rBC cores down to approximately 115 nm. As rBC cores decreases below 115 nm, thicker coatings are required to produce a measurable scattering signal. In all panels, the apparent coating thickness increases with decreasing rBC core. Below about 120 nm, the apparent rate of thickness increase with decreasing rBC is larger than above 120 nm. Since only thicker coatings can be measured

at smaller sizes, some of the increase in thickness rate is attributed to bias in the elastic scattering detection system toward thicker coatings when the rBC cores are less than 115 nm. Overall, there is some increase in coating thickness with smaller rBC core sizes, but the thicknesses represented at rBC core sizes of less than 120 nm are overestimated.

Over the rBC size range of 160-180 nm, coating thicknesses are assigned for more than

80% of rBC particles. Fig. 6 shows the average ratio of the total particle diameter (rBC core and coating) to the rBC core diameter ($D_p/D_c$) every six hours; minimum and maximum thicknesses are shown for October-November. In April 2012, there is a gradual decrease in $D_p/D_c$ from about 1.4 at the beginning to about 1.25 at the end of the month. This decrease may be representative, but a changing calibration cannot be ruled out. More variability is evident in October-November

2013, and the average minimum and maximum thicknesses are about 1.05 and 1.35; the maximum thickness starts at about 1.4 and decreases to about 1.3. Since the maximum thicknesses for October-November 2013 are close to the values for April 2012, and the calibration used to derive the minimum thickness is two years after the measurements whereas the one used for the maximum thickness occurred one year before the measurements, the true

thicknesses for October-November 2013 are likely closer to the maximum values.

The present $D_p/D_c$ can be compared with those of other studies. In background continental air over Texas and for $D_c$ of 190-210 nm, Schwartz et al. (2008) reported $D_p/D_c$ values about 1.5 times higher than reported here. A value of 2.4 at a 170 nm core size was measured in a smoke plume over Europe (Dahlkotter et al., 2014), and a study in the Finnish

Arctic found a $D_p/D_c$ of 2 for $D_c$ of 150-200 nm (Raatikainen et al., 2015). More comparable



with the present results, Sahu et al. (2012) found an average $D_p/D_c$ of 1.5 for aged biomass burning plumes and an average $D_p/D_c$ of 1.24 for aged fossil fuel combustion plumes in California, where $D_c$ was greater than 200 nm. In six Asian cities, Kondo et al. (2011) measured a median $D_p/D_c$ of 1.1 for $D_c$ of 162-180 nm near the source.

At Alert and over the scattering size range of 200-400 nm, the seasonal ratios of rBC particle numbers (incandescent events) to total particles (scattering events) are 0.12 for DJF, 0.11 for MAM, 0.16 for JJA, and 0.10 for SON. Assuming there are no smaller undetectable rBC cores in the scattering particles, the percentage of particles containing rBC are approximately between 10% and 16%, lower than the 24% found in the Finnish Arctic across the 350 nm to 450 nm size range (Raatikainen et al., 2015).

### 3.2.3 Summary

During the summer and fall seasons, number concentrations of rBC cores at Alert are a factor of 5-10 lower and exhibit a slightly smaller mode diameter than during winter-spring. For rBC cores in the 160-180 nm range, the average particle coating thicknesses in April 2012 and in October-November 2013 were estimated to range from 1.25 to 1.4. For particles scattering light equivalent to 200-400 nm PSL particles, the proportion containing detectable rBC cores is between 10% and 16%.

## 4. Discussion

### 4.1 Best estimate of aerosol black carbon at Alert

Possible reasons for the differences among the three techniques used to estimate BC at Alert (EC, EBC and rBC) are discussed in this section, leading to a best estimate for BC at Alert that may be useful for evaluation of chemical transport models.

### 4.1.1 EBC

EBC will overestimate BC if there is absorption from coexisting components and/or coatings of the rBC cores, such as brown carbon (e.g. Kirchstetter et al., 2004; Lack et al., 2013; Lack and Langridge, 2013) and fine particle dust (Weingartner et al., 2003; Müller et al., 2011). In addition, the Aethalometer response depends on filter loading and multiple scattering by the sampled aerosol particles. Recently, Backman et al. (2017) proposed a reduction of a factor of



3.2 in the light absorption coefficients derived from the Aethalometer due to multiple scattering enhancements associated with particles collected on the filter. These enhancements are considered, at least in part, in the EBC estimate by the $\alpha_{ap}$ value used with the Aethalometer, but there remains uncertainty in $\alpha_{ap}$, including the use of a constant value for all conditions.

### 4.1.2 EC

EC can be influenced by components that co-elute with oxygenated OC or brown carbon and may not be detected as rBC by the SP2 but measured by the thermal method as EC, including humic substances (natural organic material in soil and water) and humic-like substances or HULIS (e.g. Graber and Rudich, 2006) and dust. The techniques for measuring

rBC and EC examine different parts of the atmospheric BC thermal spectrum (Andreae and Gelencser, 2006): rBC is at the refractory end (3600°C), whereas EC by this thermal method is the residual part of carbon mass after heating to 900 °C, and it will include rBC and possibly some non-BC carbonaceous components that would be interpreted as BC. As shown in Fig. 8, the weekly differences in EC and rBC (EC-rBC) exhibit a moderate association with POC plus

CC (POC+CC) component of the carbonaceous aerosol.  By thermal definition, POC+CC is the carbonaceous component that eludes at $870^{o}C$ in helium (see Fig. 4), which is proportional to the amount of oxygenated OC (Chan et al., 2010) or brown carbon, and EC might not be completely separated with temperature from POC+CC component. The higher scatter in the winter (green symbol, $r^2=0.3$) and spring (red symbol, $r^2=0.4$) data could be because there are influences of

POC from multiple sources during these seasons, whereas this correlation improves during summer (blue symbol, $r^2=0.7$) suggesting that the influences on POC and EC/rBC are from a more consistent source. These strong associations among POC and EC-rBC suggest that during the temperature separation in the EC/OC thermogram, some of the POC component still remains in the EC fraction. These EC fractions, which are co-emitted with POC/BrC and likely formed

from low temperature processes (relative to $3600^{o}C$), may not be well detected as rBC by SP2. This is why EC may overestimate BC relatively to rBC. There is a reason therefore that EC may overestimate BC.

### 4.1.3 rBC

The rBC masses are derived under the assumptions that the calibration curve can be extrapolated linearly above 333 nm (including calibrations for #17, not shown in Fig. 2) and that the



distributions of core diameters outside of the measured size region (80-530 nm) are represented by a log-normal function. The linear extrapolations of the calibration curves (e.g. Fig. 2) offer no suggestion of a bias. The distributions in Fig. 3 suggest that most of the rBC mass is accounted for within the above measured size range, and that the log-normal approximation is reasonable.

The density estimate of the particles used in the calibration of the SP2 is a potential source of bias in the rBC estimate. The most up-to-date and experimentally-derived parameterization (Gysel et al., 2011) has been used here, but if the density assumption of the calibration particles differs from the "true" calibration particle density, the rBC mass concentrations will be biased. It cannot be also ruled out that Aquadag correction could also introduce some bias.

### 4.1.4 BC mass best estimate

At Alert, the absolute concentrations of EBC, EC, and rBC are each relatively small, but both EBC and EC are biased high relative to rBC. As above, there are valid reasons to expect those high biases.  A clear bias in the rBC measurement cannot be identified, but neither can it be

ruled out. The rBC mass concentrations will also be biased relative to true BC: rBC satisfies most of the five characteristics representing BC discussed by Petzold et al. (2013), but there may be some limitations as it pertains to their morphology criterion and the technique offers no guarantee that incandescing components are completely insoluble. Considering all arguments, including EC and rBC being more specific measurements, our best estimate of BC at Alert, to be

used for comparison with chemical transport models, is an average of the rBC and EC measurements with a range bounded by the rBC and EC and their combined measurement uncertainties, respectively.  Thus, the best estimates of winter-, spring-, summer-, and fall-averaged BC with atmospheric variability at Alert for this study period are $49\pm28$ ng m$^{-3}$, $30\pm26$ ng m$^{-3}$, $22\pm13$ ng m$^{-3}$, and $29\pm9$ ng m$^{-3}$, respectively.

### *4.2 Seasonal variability of MAC*

The mass absorption coefficient (MAC) is used to derive a mass concentration from a particle light absorption measurement. For BC in a freshly emitted aerosol, MAC has been estimated to be $7.5\pm1.2$ m$^2$g$^{-1}$ at a wavelength of 550 nm (Bond et al., 2013). The MAC value

will vary in time and space depending upon source emissions and transformation during transport as the particles age (Chan et al., 2011; Bond et al., 2013). In general, MAC increases as





more material coats the BC as discussed in section 3.2.2. Other important components of the aerosol that absorb visible light tend to have much weaker absorption efficiencies at visible wavelengths; approximately 0.009 $m^2g^{-1}$ at 550 nm for dust (Petzold et al., 2009), and approximately 1 $m^2g^{-1}$ at 550 nm for brown carbon (Kirchstetter et al., 2004; Chakrabarty et al.,

2010). Uncertainty in the MAC value for BC is associated with both the absorption measurement and the BC mass concentration measurement.

       The estimated MAC values are illustrated in Fig. 9, where $\sigma_{ap}$ values are plotted against our best estimate for the BC mass concentrations (i.e. the average of EC and rBC) for the spring (MAM) and winter (DJF) periods. In each plot, the black points represent all available data.

Those data are scrutinized in two ways. First, to help reduce uncertainty in the mass concentration estimate, observations are excluded from the analysis if the magnitude of the difference between EC and rBC relative to the mass concentration estimate is less than 75%. This is an arbitrary constraint, but using 50% or 100% offers relatively small changes. For example, in the spring case, the slope and intercept of the red circles are 0.0080 and 0.22,

respectively for a constraint of 50% and 0.0071 and 0.21, respectively, for a constraint of 100%. The impact on the winter results is less due to the higher correlations. Evident in Fig. 9, the overall effect of this constraint is to reduce the impact of lower $\sigma_{ap}$ and mass concentration points, which have greater relative uncertainty. Second, the $\sigma_{ap}$ values are constrained to be greater than or equal to 0.2 $Mm^{-1}$. This is done to help further reduce relative uncertainty

associated with low $\sigma_{ap}$ values. In each of the spring and winter cases, this constraint removes only one datum: other such points are removed by the first constraint.

       The mass concentration estimate is for BC, and if the measured absorption is due to BC only then a best fit should go through the origin. The intercept could be a result of incomplete corrections for artifacts in the $\sigma_{ap}$ from the PSAP, it may represent the mean of other absorbing

species or a combination of those two. With the intercept subtracted from the scrutinized data, the final curve (black crosses) represents our best estimate for light absorption as a function of BC mass concentration. Scatter in the data may also be due to either incomplete artifact corrections or variations in other light absorbing components of the particles. The greater scatter in the spring data compared with winter may be consistent with an increased presence of brown

carbon during spring, since OM at Alert is a factor of two higher in the spring than during winter (Leaitch et al., in preparation). There are 10 data points for the summer period (JJA), but none of





them fall within the above constraints, largely due to the low mass concentrations and values of $\sigma_{ap}$. For the fall, there are a total of five data points, two of which are constrained as above and both of which yield a MAC value of 13.4 after subtraction of a positive intercept of 0.02 Mm$^{-1}$. Reasons for the two relatively high fall values of MAC are unknown, but the spring and winter

data offer larger datasets and consistent average MAC values: 7.6±3.8 m$^2$ g$^{-1}$ for spring and 7.7±3.8 m$^2$ g$^{-1}$ for winter. These MAC values for spring and winter are reported in Table 2.

There are only five one-week averages (during April 2012 and November 2013) with corresponding MAC values and $D_p/D_c$ values from the coating analysis. Those MAC estimates, based on the average of rBC and EC and following the above criteria, for 550 nm wavelength are

plotted against weekly-averaged $D_p/D_c$, as shown in Fig. 10. Also shown in Fig. 10 is the variation in MAC for coating thickness expected from the core-shell model of Shiraiwa et al. (2010) starting with the MAC value for uncoated BC from Bond and Bergstrom (2006) of 7.5±1.2 m$^2$ g$^{-1}$. The present observations indicate a significant increase in MAC with increased coating thickness ($r^2$=0.3, p<0.001), and the slope of the curve over the range of observations is

steeper than the core-shell theoretical curve. The core-shell curve falls within the uncertainty of the regression curve at 880 nm, where BC is the dominant absorber, and therefore these results cannot be interpreted as indicating stronger absorption than expected from the core-shell model. However, the shell-core model is an ideal representation, and enhancements of 50% or more in absorption are possible due to presence of black carbon aggregates as opposed to simple

spherical cores (Bond et al., 2013). High relative humidity has been found to amplify absorption by as much as a factor of 2.7 (Brem et al., 2012), but the RH in the sampling lines at Alert is <40% and it is unlikely to be a significant influence here.

### 4.3 Comparisons with other studies

Ground-based measurements at other Arctic sites have also provided comparisons of

various BC techniques. Eleftheriadis et al. (2009) found EBC and EC were comparable at Ny-Ålesund during July 1998 to August 1999. Raatikainen et al. (2015) showed comparisons among the SP2, the Aethalometer and the Multi-angle Absorption Photometer (MAAP) measurements over two-month period (Dec 2011 to Jan 2012) in northern Finland. Their mean rBC estimate was 27 ng m$^{-3}$ integrated over 75 to 655 nm sizes, compared with 38 ng m$^{-3}$ integrated over 75 to

1000 nm for the same time period at Alert, and their average mean rBC diameter of 194 nm is





the same as for Alert (194±17 nm). However, their regressions of EBC to rBC gave a slope of 4.3 compared with a slope of 1.6 for Alert.  Alert, located at 82.5$^o$ N, a general circulation brings more direct transport from Siberia during this time period (Sharma et al., 2006), whereas the Finnish site, located at 67$^o$ N, was influenced more by European sources. Massling et al. (2015)

showed comparisons between EBC_aeth and EC at the Villium Research Station, Station Nord, Greenland (81$^o$N; ca. 700 km from Alert) for 2011-2013. A MAAP was used to measure EBC at 670 nm wavelength and a MAC value of 6.6 m$^2$ g$^{-1}$ (the default for the MAAP) was used to convert absorption from MAAP to EBC mass concentrations. EC was determined using thermal analysis following the European Supersites for Atmospheric Aerosol Research (EUSAAR-2

protocol; Cavalli et al., 2010). The seasonal values of EBC are strikingly similar for Alert and Station Nord: respectively, winter 62 ng m$^{-3}$ and 67 ng m$^{-3}$; spring 57 ng m$^{-3}$ and 54 ng m$^{-3}$; summer 13 ng m$^{-3}$ and 11 ng m$^{-3}$; fall 19 ng m$^{-3}$ and 22 ng m$^{-3}$. They report EBC mass concentrations a factor of two higher than the EC mass concentrations.

Kondo et al., 2011 conducted a study in six cities in Asia by heating the sample to 300$^o$C

to burn off the organics (HULIS by 30%) and lower the potential artificial enhancement in absorption by non-refractory compounds. They measured the absorption with a variant of the PSAP (COSMOS). They obtained MAC_rBC and the MAC_EC values of 5.5 and 5.4 m$^2$g$^{-1}$, respectively. The same comparison between PSAP and COSMOS absorption measurements showed 22% lower COSMOS absorption at Barrow, Alaska (Sinha et al., 2017). Although the

two methods are different in concepts, the subtraction of the intercept in the plots in Fig. 9 reduces the PSAP value by an average of 50% for the spring and 25% for the winter. This gives a MAC of 7.6±3.8 and 7.7±3.8 for winter and spring which is similar to results of 9.0 m$^2$g$^{-1}$ obtained by Sinha et al. (2017) at Barrow for 2012 and 2013 time-period.

The agreement improved between the weekly averages of uncorrected EBC from

Aethalometer and the best estimate of black carbon mass by using the best estimate of black carbon instead of rBC or EC masses alone at Alert (Supplement_Fig.), red and green triangles; slope=1, r$^2$=0.9 and slope=0.9, r$^2$=0.9 for winter and spring and r$^2$=0.9) increasing confidence in the optically based mass measurements at Alert as trends have been drawn from these optical measurements (Sharma et al., 2013).




## 4. Summary and Conclusions

Estimates of BC at Alert, Nunavut over a two year and nine month period (Mar 2011 to Dec 2013) are made based on three different techniques: an Aethalometer for EBC, analysis of thermally evolved carbon from weekly quartz filters for EC, and an SP2 for rBC. Over the study period, on average both EBC and EC are 2.7 and 3 times higher than rBC respectively. EBC is biased higher than EC for the months of higher concentrations (November to May) and EC is biased higher than EBC during the lower-concentration summer months (June-August). The winter-spring EBC bias is attributed to the presence of absorbing substances other than BC and by scattering associated with particles accumulating on the filter that can enhance absorption by the BC relative to the atmosphere. Uncertainties also exist in the specific attenuation coefficient used to convert Aethalometer light attenuation to EBC, but it is unclear if or how that may bias the EBC mass estimate.

EC and rBC differ in that EC, as measured here, is the carbon that evolves to $CO_2$ after heating at $900^{\circ}C$ in an oxygen-rich atmosphere, and rBC is based on the incandescent signal from particles heated to approximately $3600^{\circ}C$. Those rather substantial differences and an observed association of the difference between EC and rBC with pyrolysis OC plus carbonate carbon suggest the present EC is likely biased high by some pyrolyzed OC. The calibration procedure is a possible source of a bias in the rBC measurements, but the magnitude and direction of a possible bias is not clear. The definition of rBC is a potential bias as it relates to BC. Refractory BC satisfies most of the five characteristics representing BC given by Petzold et al. (2013), but there may be some limitations as it pertains to their morphology criterion and the technique offers no guarantee that incandescing components are completely insoluble. This argument also applies to EBC and EC.

Our present best estimate of BC at Alert, offered for possible use in model evaluation, is an average of the rBC and EC measurements with a range bounded by the combined rBC and EC measurement uncertainties. For this study period, the best estimate averaged BC at Alert with atmospheric standard deviation for the winter, spring, summer and fall are $49\pm28$ ng m$^{-3}$, $30\pm26$ ng m$^{-3}$, $22\pm13$ ng m$^{-3}$, and $29\pm9$ ng m$^{-3}$, respectively. The propagated uncertainties in the averaged mass during spring and winter are $\pm30\%$ but this uncertainty increased during summer and fall to around $\pm40\%$.





During summer and fall, the number concentrations of particles with detectable rBC are 5-10 times lower than during winter-spring, and exhibit a slightly smaller mode diameter. For rBC cores in the 160-180 nm range, the average ratio of total particle diameter to rBC core diameter ($D_p/D_c$) was measured for April 2012 and October-November 2013 and was found to

range from 1.25-1.4. For particles scattering light equivalent to 200-400 nm PSL spheres, the fraction containing rBC cores is estimated to be between 10% and 16%, but smaller undetectable rBC cores (< 80 nm) in some of the scattering particles cannot be excluded.

Light absorption measured with a PSAP was used with the EC and rBC averages to calculate the MAC value at 550 nm wavelength ± uncertainty are 8±4, 8±4, 5±2.5, and 9±4.5 $m^2$

$g^{-1}$, for winter, spring, summer, and fall respectively. These values are further refined by adjusting the absorption to only black carbon and obtained winter and spring MAC of 7.6±3.8 and 7.7±3.8 $m^2g^{-1}$. Only winter and spring estimates of MAC were possible due to the low number of usable data points available from the summer and fall periods.

**Acknowledgements**

The authors would like to acknowledge the Department of National Defense and Environment and Climate Change Canada for operating the Dr. Neil Trivett Global Atmosphere Watch Observatory at Alert, NU. The authors would like to thank the operators and students for day-to-

day instrument maintenance and activities at the Observatory. We are particularly grateful to Desiree Toom for providing the assistance with the calibration systems and other aspects of the aerosol measurements, and to our Arctic Coordinator, Andrew Platt, for his continuing support of our activities at Alert. Thanks to Vince Vetro for averaging of high time resolution EBC and rBC data to the EC filter times. Allan Bertram and Sarah Hanna from University of British

Columbia were supported by the Natural Science and Engineering Research Council of Canada through a Climate Change and Atmospheric Research Grant (NETCARE).





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



| Instrument type | Method | Model # Detection range | Manufacturer | Measurement | Size-cut | Measurement Time-period |
|---|---|---|---|---|---|---|
| PSAP | Optical | ------ | Radiance Inc. | Light absorption, $\sigma_{ap}$ | 1 μm impactor | 8 Mar, 2011 to 31 Dec, 2013 |
| OC/EC Analyzer (Lab mode) | EnCan-total-900, a Thermal Protocol | ------ | Sunset Lab Inc. | Elemental Carbon/Organic Carbon, EC/OC | 1 μm cyclone | 8 Mar, 2011 to 31 Dec, 2013 |
| Aethalometer | Optical Absorption converted to EBC using $\alpha_{ap}$=16.6 m$^2$g$^{-1}$ | AE-31 | Magee Inc. | Equivalent Black Carbon, EBC | TSP since 1989 | 8 Mar, 2011 to 31 Dec, 2013 |
| SP2 | Incandescence | SP2-C #17, 4 channels 65-225 nm VED* | Droplet Measurement Technology Inc. | Refractory black carbon, rBC  -Mass integrated over 80 to 1000 nm size range | 1 μm cyclone | 8 Mar, 2011 to 24 Mar 2012 |
| | | SP2-C*, #44, 8 channels 65-530 nm VED | | | | 27 Mar, 2012 to 22 Sept, 2013 |
| | | SP2-D, #58, 8 channels 80-530 nm VED | | | | 27 Sept, 2013 to 31 Dec, 2013 |

**Table 1: Various instrument used in this comparison study.**

**\*VED =volume equivalent diameter**





| Season | EC, ng m$^{-3}$ Avg. (25th, 75th) | rBC, ng m$^{-3}$ Avg. (25th, 75th) | EBC_aeth, ng m$^{-3}$ Avg. (25th, 75th) | (EC+rBC)/2 ng m$^{-3}$ | EC vs rBC Lin. reg., r$^2$ (p) | EBC vs rBC Lin. reg., r$^2$ (p) | EBC vs EC Lin. reg., r$^2$ (p) | $\sigma_{ap}$, Mm$^{-1}$ @ 550 nm Avg. ± std. dev. | MAC[(EC+rBC)/2] (m$^2$g$^{-1}$) Lin. reg ± uncer. (r$^2$) MAC[(EC+rBC)/2] (m$^2$g$^{-1}$) (Intercept subtracted) |
|---|---|---|---|---|---|---|---|---|---|
| Winter (DJF) | 48 (17, 63) | 33 (15, 40) | 62 (31, 68) | 49 (22, 71) | 1.7, 0.8 (0.07) | 1.4, 0.9 (0.002) | 0.64, 0.7 (0.14) | 0.46±0.3 | 8±4 (0.92) 7.7±4 (0.8) |
| Spring (MAM) | 43 (23, 56) | 25 (13, 28) | 57 (37, 71) | 30 (24, 44) | 1.3, 0.7 (0.002) | 1.2, 0.8 (6.1e$^{-28}$) | 0.63, 0.5 (0.017) | 0.45±0.25 | 8±4 (0.81) 7.6±4 (0.9) |
| Summer (JJA) | 19 (4, 27) | 6 (3, 8) | 13 (7, 15) | 22 (12, 32) | 3.5, 0.5 (3.4e$^{-5}$) | 1.4, 0.6 (2e$^{-5}$) | 0.24, 0.4 (0.03) | 0.10±0.07 | 5±2.5 (0.6) NA |
| Fall (SON) | 13 (3, 17) | 8 (2, 10) | 19 (6, 25) | 29 (16, 46) | 1.3, 0.8 (0.043) | 1.6, 0.8 (4.5e$^{-3}$) | 1, 0.75 (0.06) | 0.15±0.16 | 9±3.4 (0.5) NA |

**Table 2**: Seasonal averages** of EC, rBC, EBC_aeth, and $\sigma_{ap}$ at 550 nm and interquartile difference reported at 25 and 75 percentile. For EBC_aeth, a default $\alpha_{ap}$ value of 16.6 m$^2$g$^{-1}$ was utilized at 880 nm. The rBC mass was integrated over 8-1000 nm sizes. Linear regressions among techniques were performed using least squares method, r$^2$ is the coefficient of determination and averages for pair of techniques are statistically different for p<0.01. The mass absorption cross section (MAC) was derived from a linear regression between $\sigma_{ap}$ at 550 nm and (EC+rBC)/2 masses. The second MAC value was determined by scrutinizing data by subtracting off the intercept value from $\sigma_{ap}$ (see text for details in section 4.2, Fig. 9). A propagated uncertainty of 50% was determined for MAC from individual uncertainties in $\sigma_{ap}$ (20%) and (EC+rBC)/2(30%).

**seasonally averaged values included all negative $\sigma_{ap}$ and EBC concentrations and averaged data was considered valid when more than 50% of values were present in the averaging period.





| | All data Conc. (ng m$^{-3}$) | | All data Conc. (ng m$^{-3}$) (above det. lim. values) | | Conc. (ng m$^{-3}$) and ratios Pairwise (only include data when both pairs of data are available and above det. lim.) | | | | | |
|---|---|---|---|---|---|---|---|---|---|---|
| | | | | | EBC vs rBC | | EC vs rBC | | EC vs EBC | |
| | Mean | Median | Mean | Median | Mean | Median | Mean | Median | Mean | Median |
| EBC | 34±31 | 23 | 36±30 | 25 | 37±31 | 26 | --- | | 43±31 | 32 |
| EC | 30±32 | 18 | 40±32 | 32 | --- | | 40±32 | 32 | 43±33 | 38 |
| rBC | 17±19 | 11 | 17±19 | 11 | 18±20 | 11 | 21±20 | 14 | --- | |
| EBC/rBC | --- | --- | 2.7±1.5 | 2.2 | 2.7±1.5 | 2.2 | --- | | --- | |
| EC/rBC | --- | --- | 3.1±2.6 | 2.2 | --- | | 3.1±2.6 | 2.3 | --- | |
| EC/EBC | --- | --- | 1.2±0.8 | 1.0 | --- | | --- | | 1.2±0.8 | 1.0 |

**Table 3:** Statistical parameters such as mean, median and standard deviation for all data, Mar 2011 to Dec 2013 and data with only above detection limit values included. The ratios are only meaningful for data above the detection limit values. Also pairwise statistics available for data
5    set when both pairs in comparison had data.





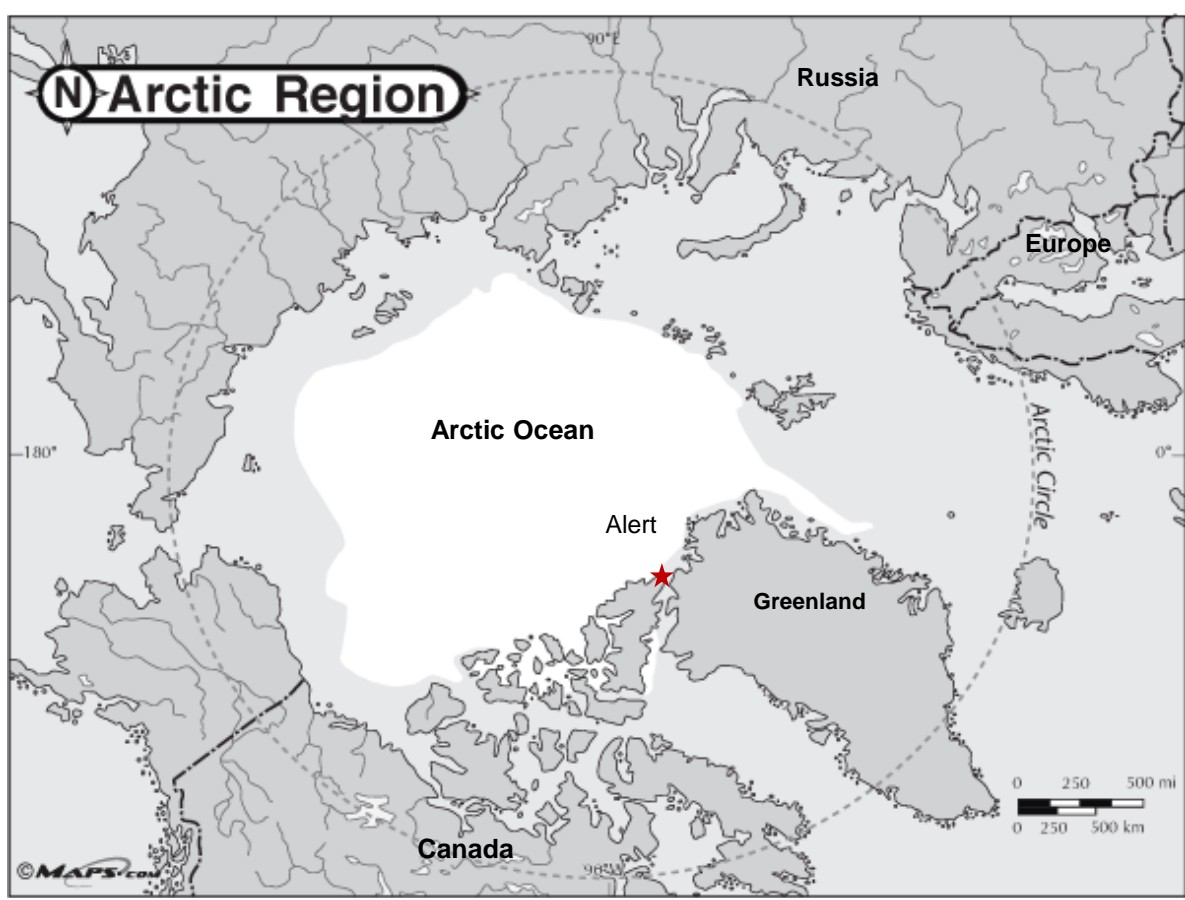

Figure 1: Alert (red star) is located at the northwest tip of Ellesmere Island in Nunavut,
Canada at 82°N and 62.5°W at an altitude of 186 m ASL.



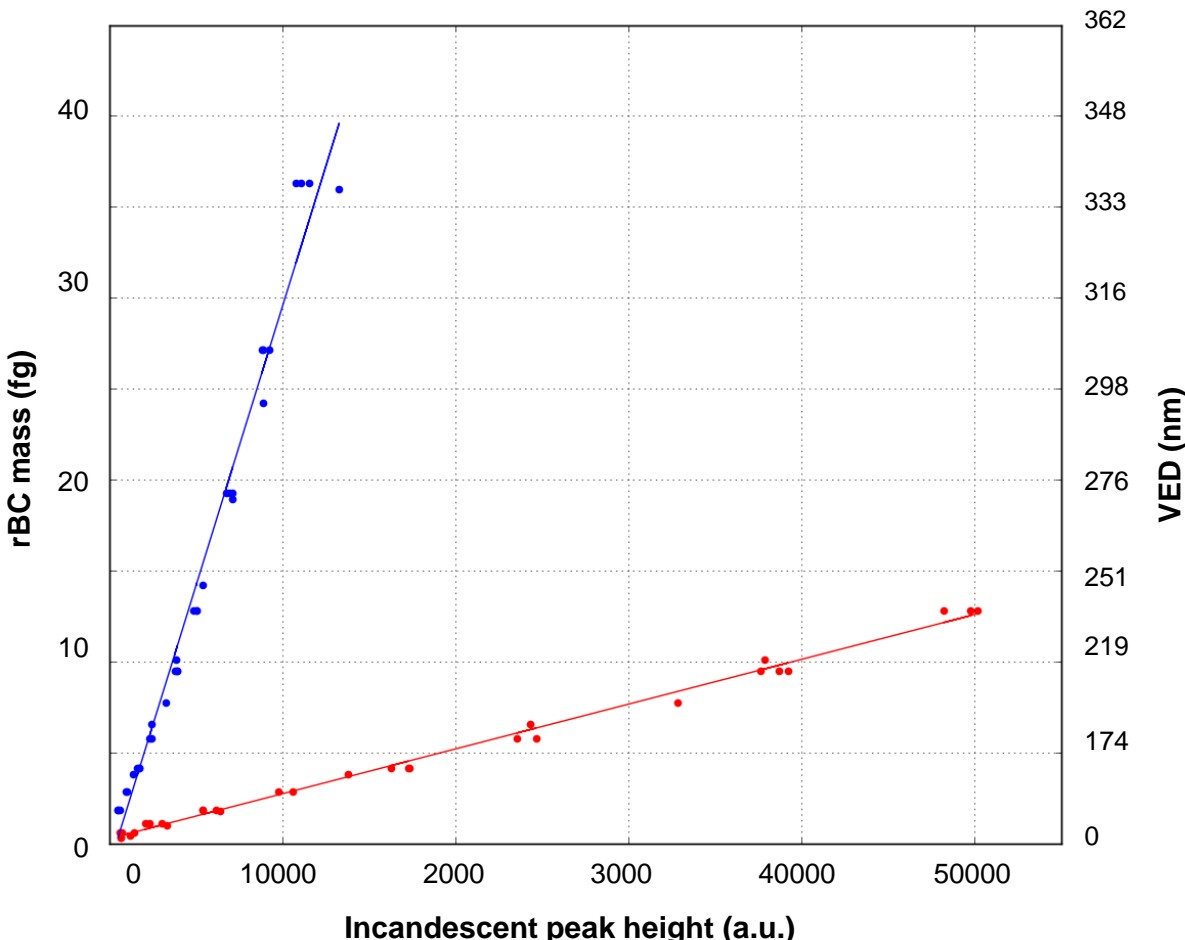

Figure 2: A calibration curve for SP2 #58 shows an incandescent peak height
with an arbitrary unit response to rBC mass (fg). The data points in blue circles are response to
low gain channel and in red circles are SP2 response to high gain channel. The right hand axis
shows volume equivalent diameter (VED) in nm.





Figure 3: Seasonally-averaged mass (a) and number (b) distributions, by Volume Equivalent Diameter (VED), of rBC particles detected by the SP2 at Alert, NU from March 2011 to December 2013. The number distributions are estimated from the fits to the mass distribution. The black, green, blue, and red data curves represent the three-month average season: winter (DJF), spring (MAM), summer (JJA), and fall (SON), respectively. On average, a modal mass diameter at 225 nm was measured for rBC during winter and around 170 nm during summer. The number distribution shows mode at 100 nm during winter due to long range transport influence and during the light period in the spring, giving smaller particle diameter VED indicates volume equivalent diameter. The solid circles represents size range of calibration.



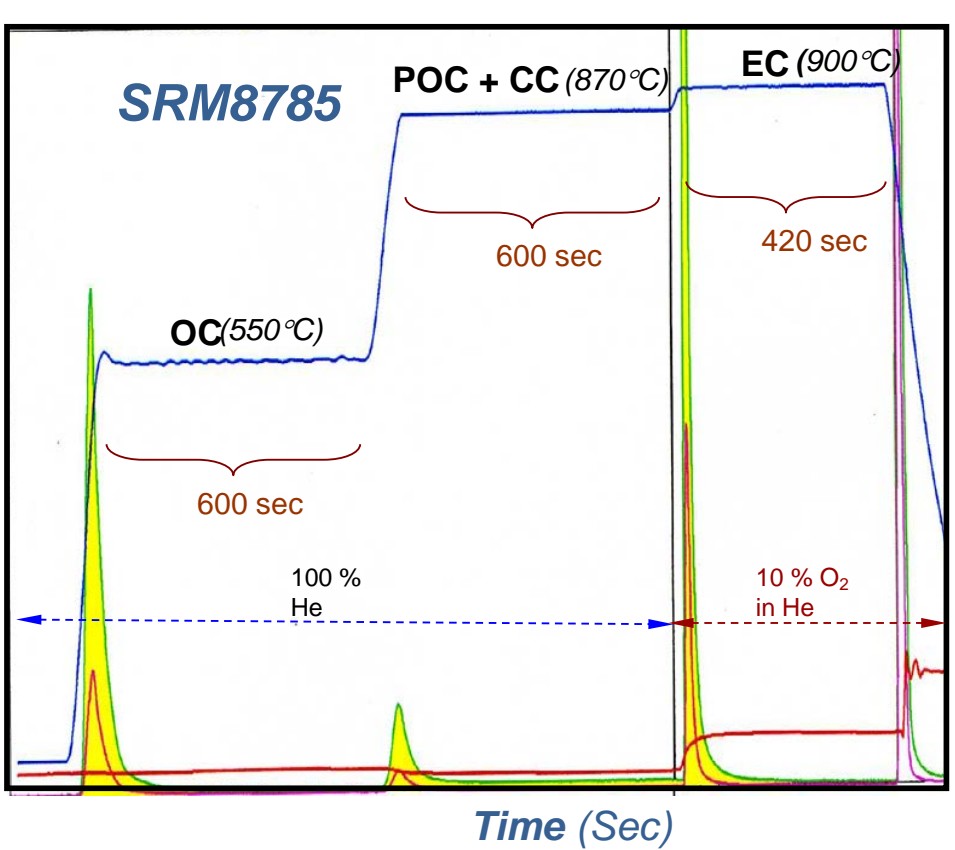

Figure 4: A typical thermograph of the NIST standard (SRM8785: urban dust on filter) using the thermal method (EnCan-total-900). The solid dark blue line indicates the temperature time series. The green and red peaks highlighted in yellow are two Flame Ionization Detector (FID) signals (one with higher sensitivity than the other). The solid red line is the laser transmittance; however it is not relevant here since transmittance is not used to determine OC and EC concentration. The determinations of OC, POC + CC, EC mass fractions are purely based on temperature and retention time.



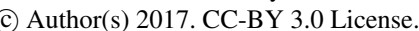

Figure 5: The temporal variation of BC at Alert, NU from March 2011 to December 2013: (a) hourly refractory BC, (b) hourly equivalent BC from Aethalometer @ 880 nm, (c) weekly EC concentrations, and (d) monthly averaged mass and of each technique and standard deviation.





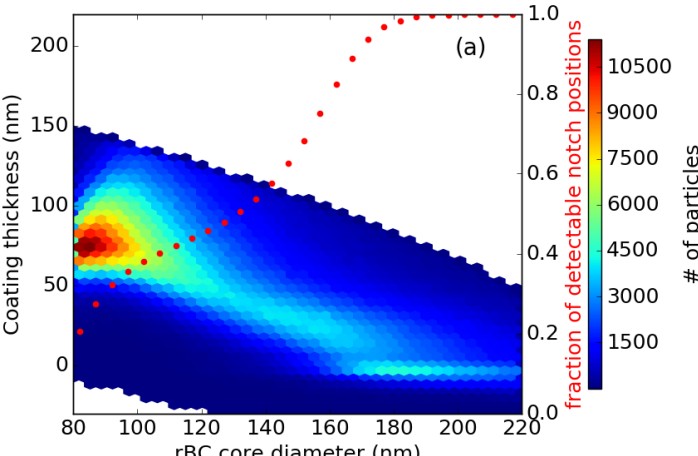

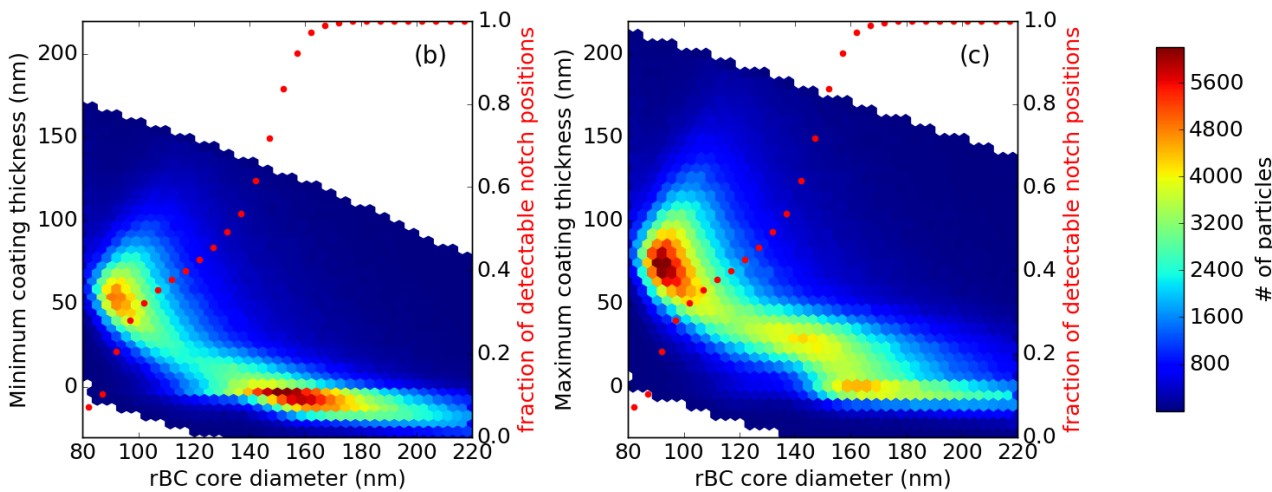

Figure 6: Two-dimensional histograms of coating thickness as a function of core size for rBC particles at Alert, NU: (a) the average coating thickness during April 2012, (b) the minimum coating thickness during October and November 2013, and (c) the maximum minimum coating thickness during October and November 2013. The red circles indicate what fraction of rBC cores at each size could be successfully assigned a coating thickness.



Figure 7: Time series of the mean ratio of total particle diameter (core and coating, Dp) to rBC core diameter (Dc) for (a) April 2012 and (b) October-November 2013 with the 25th and the 75th percentile represented by the shaded region. The minimum and maximum range in the Dp/Dc are shown for the October-November 2013 period because the sensitivity of the scattering channel changed, resulting in two valid calibrations.




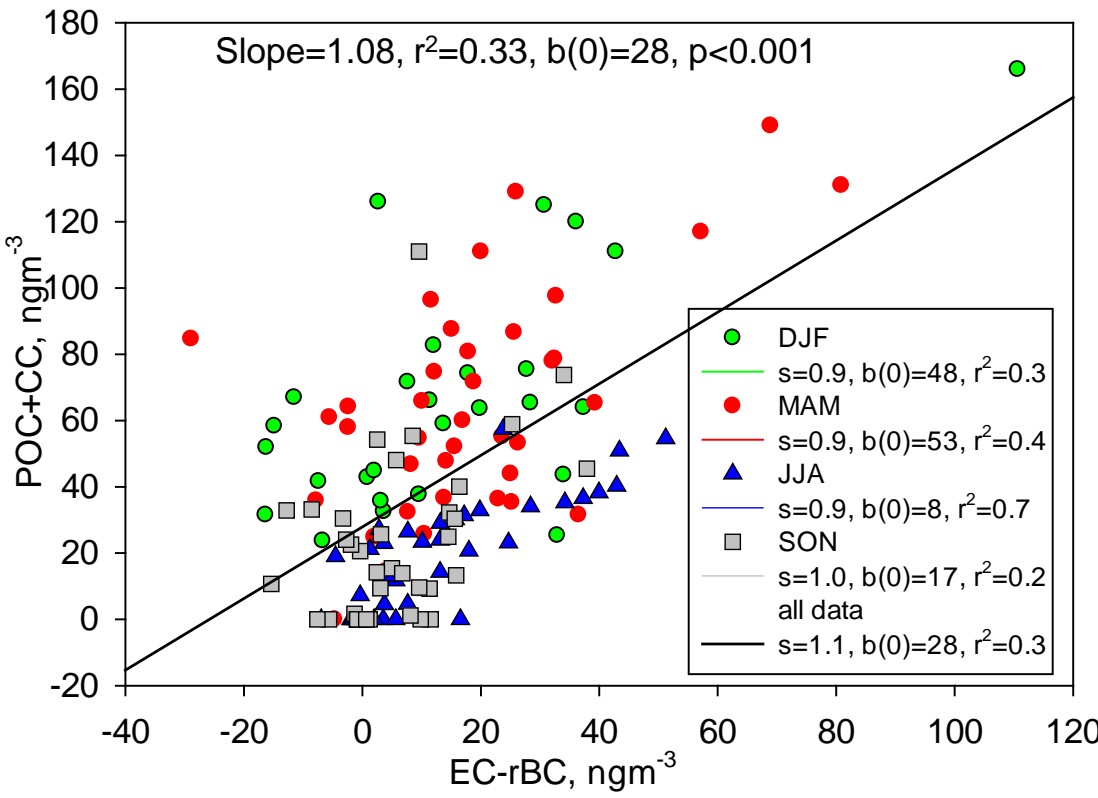

Figure 8: A weak but significant (p<0.001) correlation among the difference between two weekly masses (EC and rBC) at Alert and the oxygenated carbon component as represented by POC+CC, show some positive tendency indicating incomplete separation of EC from other co-eluding thermal components. The correlation improves for summer (blue symbol, $r^2$=0.7) indicating a more consistent oxygenated source as compared to other seasons where $r^2$=0.3.






Figure 9: Light absorption coefficient ($\sigma_{ap}$) at 550 nm versus the mass concentration of the average of EC and rBC for winter (a) and spring (b). Each plot shows all available data (black dots: 31 spring points and 21 winter points), a subset of all data scrutinized for closer agreement between EC and rBC and higher $\sigma_{ap}$ (red circles:18 spring points; 16 winter points) and the scrutinized subset with the intercept subtracted (black crosses). See text for discussion of the rationale for scrutinization.




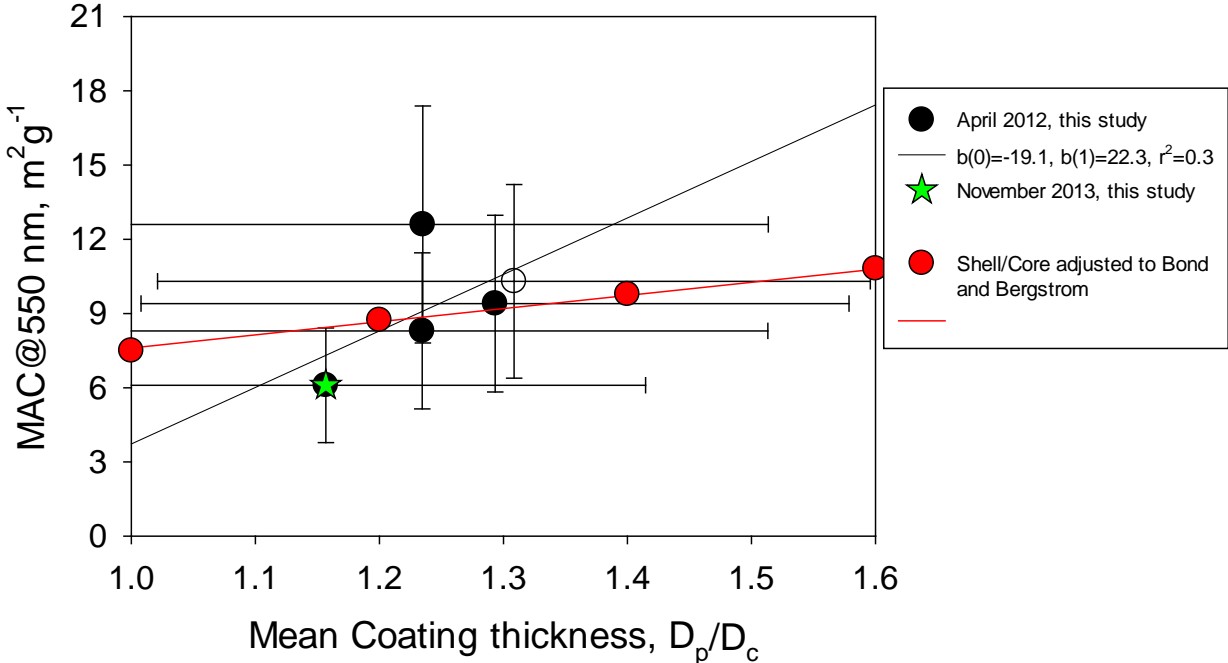

Figure 10: The comparison between the MAC and weekly averaged mean coating thickness
shows a positive and significant relationship (p<0.001). Aerosols seem to have thicker
coatings in April (black circle) than November (green star), also note that only values above the
detection limits for MAC were considered. Red circles are adjusted Shell/core to Bond and
Bergstrom, 2006 for uncoated particles at 7.5 m$^2$g$^{-1}$.