# Peer review of "An Evaluation of three methods for measuring black carbon at Alert, Canada"

_Atmospheric Chemistry and Physics, 2017_

## Referee Comment (RC1) · Anonymous Referee #2 · 16 Jun 2017

GENERAL

The paper presents a comparison of methods for measuring BC in the high Arctic and the retrieved estimation of the MAC of BC. The methods are completely different: filter-based attenuation (aethalometer and PSAP), incandescence (SP2) and thermal desorption (EC) so they complement each other. The paper is well written and it discusses also the uncertainties in a detail so it is an important and useful work. I found some points in it, however, that I wish the authors would address before publishing it in ACP. The most important question is that there is no scattering correction for the Aethalometer data. Was the PSAP data corrected for scattering according to Eq. (4)? If so, you must have scattering data. If not, absorption coefficients from the PSAP are overestimated. It has been well known for a long time that both the aethalometer and

the PSAP give a signal even for purely scattering aerosol, so called apparent absorption (eg., Arnott et al. 2005, Collaud Coen et al., 2010). When you are measuring aerosol at high SSA and low absorption coefficient, this effect definitely becomes important. It may sometimes even be possible that the whole absorption is apparent and thus also EBC. The high EBC-to-rBC ratio might be explained by this kind of an artifact. Did you have a nephelometer there or could you estimate scattering by any method? You should discuss this somehow.

Detailed comments

P1, L3 use the same amount of significant nmbers

P3, L9 " The light absorption is converted ..." No, the aethalometer converts attenuation coefficient into EBC.

P6, L14. What is ECCC?

P9, L17: dATN cannot be > 2 if you use the definition of ATN in Eq. (1) – the manufactuter presents TN in percents.

P10, L9 " Equation 7 is rewritten in Eq. 6" Should this be "Equation 4 is rewritten in Eq. 7"?

P1, L20, Eq. 9. I don't understand how you get the numbers 0.02, 1.44 and 0.24. Please explain.

P10, L25, Weekly zeroes – how long time?

P11, L1-17. The uncertainty analysis of the SP2 is very short compared with that if the PSAP. The numbers are simply given. Can you give any more details? Is, for instance the number 19% the uncertainty of the slope?

P13, L13-14. "... a factor of 3 higher for all data..." Why is this ratio so different from the slopes ~1-2 presented in Table 2?

[Figure]

P15, L8, L10. What is "rate of thickness increase" and "thickness rate"?

P16, L6. What do you mean by "events"?

P16, L12-L17. Also somewhere earlier: could you estimate the mass or volume fraction of BC in the size range where you do have the coating data?

P16, Section 4.1.1 EBC. Here you should give some kind of an estimate of uncertainty due to scattering. See my general comments.

P17-18,Section 4.1.3 rBC You fit a single lognormal mode. Fine, I would also. However, do you have any info on whether there might be larger, coated BC particles?

P21, L7. MAAP measures at 637 nm, see Müller et al.

Table 2. In some columns there is lin.reg. Are the associated numbers slopes? Why don't you give the slope and offsets and their uncertainties – regression codes give them.

Fig 6. The y axes start from < 0 which implies negative thickness. Can't be true.

Fig 8. In the legend there are several lines but in the figure only one...
* * *

---

## Referee Comment (RC2) · D. Baumgardner (Referee) · 20 Jul 2017

This evaluation of three measurement techniques at a remote monitoring site provides a useful, quantitative comparison of three, distinctly different approaches for deriving mass concentrations of black carbon. The paper is well organized and addresses the uncertainties and limitations of the three techniques that yield EBC, EC and rBC.

I have few questions, comments or concerns. Those that I do have are listed below; however, I would offer the authors one suggestion that might provide additional insight into the variations that are seen in the MAC and differences between EBC and EC. Given that multi-wavelength sensors were implemented, the Angstrom Absorption Exponent (AAE) can be derived that has been shown to be sensitive to the source of

combustion that produces the BC. Perhaps the AAE could be calculated and drawn on Fig. 5d along with the values of EBC,EC and rBC, or related to the MAC and rBC coating to link these variations to the combustion type?

Additional comments

Page 6, Line 7: "There are no other scattering or absorption corrections...". I don't understand why corrections are not being applied when further on, the PSAP is corrected.

Page 6, line 28: How were PSAP measurements converted from 530nm to 550 nm?

Section 2.4: The uncertainty estimates should be added in Table I.

Section 4.1.4: Should explain why EBC is not used in the best estimate of BC.

Page 21, Supplement figure. The caption refers to green "stars" that are actually diamonds. In the text they are referred to correctly.

---

## Author Comment (AC1) · 7 Sep 2017

Reply to reviewer #2, anonymous is as follows on "An Evaluation of three methods for measuring black carbon at Alert, Canada" by Sangeeta Sharma et al.

We would like to thank reviewer #2 for accepting to review this manuscript and providing such positive and constructive remarks. One major concern reviewer #2 has is that why was the scattering correction not applied to Aethalometer data. Our main purpose for using the Aethalometer data in this paper is to assess how unmodified EBC compares to other more absolute mass techniques for measurement of "black carbon". That's the reason we used unmodified EBC. We also realize importance of making scattering correction to that data but only if absorption measurements are used. There

is enhancement in the absorption in Aethalometer measurements due to filter matrix as well as scattering components on the aerosol and it could be as much as by a factor of 3.2 (which includes loading and scattering) as recently estimated by Backman et al., 2017 for several Arctic locations including Alert. Magee uses a much higher $\alpha$ap value than needed for aerosol in the atmosphere. A $\alpha$ap value of 16.6 m2g-1 at 880 nm has been used in the Aethalometer firmware by the manufacturer to compensate for these artifacts and according to the manufacturer gives the best estimate of EBC. We wanted to demonstrate how well Aethalometer is estimating EBC with Magee's $\alpha$ap value at 880 nm as we have long term trends in the EBC measurements derived from Aethalometer at Alert. In the past, we have compared EBC to EC to apply a correction to EBC at Alert and the two techniques agree quite well during the Arctic Haze time. Modified by adding these lines on:

P6, lines 8-11: "There are no other scattering or loading corrections applied to Aethalometer data because a comparison of unmodified EBC mass to best estimate of "BC" mass values are also made in this paper. And that how well an enhancement in the absorption due to total scattering (loading and scattering by aerosol) has been compensated by using a higher $\alpha$ap value by the Aethalometer firmware. This is further discussed in section 4.1.1."

Also section 4.1.1 explains this well on P17 lines15 to 28: "EBC will overestimate BC if there is absorption from coexisting components and/or coatings of the rBC cores, such as brown carbon (e.g. Kirchstetter et al., 2004; Lack et al., 2013; Lack and Langridge, 2013) and fine particle dust (Weingartner et al., 2003; Müller et al., 2011). However, the influence of brown carbon may be minimal at Alert as values of the Angstrom Absorption Exponent (AAE) are between 0.5 and 1.5 suggesting predominantly fossil fuel influence (see Supplemental section and Figure_Supplemental_1b). In addition, the Aethalometer response depends on filter loading and multiple scattering by the filter medium and sampled aerosol particles. Scattering correction thus becomes important in cases when the aerosol has higher scattering with respect to total extinction (absorption+scattering), i.e. absorption is low. This is not the case at Alert especially during the Arctic haze time. Summertime measurements could fall into this scenario. Recently, Backman et al. (2017) proposed a reduction of a factor of 3.2 in the light absorption coefficients derived from the Aethalometer due to multiple scattering enhancements associated with particles collected on the filter. These enhancements are considered, at least in part, in the EBC estimate by the ïĄ̨ap value used with the Aethalometer, but there remains uncertainty in ïĄ̨ap, including the use of a constant value for all conditions. EBC (unmodified) needs to be evaluated due to these reasons in comparison to absolute measurements mass techniques."

PSAP absorption coefficients have been corrected by using scattering data measured by a 3-w TSI Nephelometer. The main purpose for applying correction for PSAP absorption was to derive MAC values at this location by using best estimate of averaged mass of EC and rBC measurements. The scattering correction was absolutely necessary for this purpose for the PSAP absorption.

Added on P6, lines 30, P7 lines1-4: "Aerosol light scattering, ïĄ̧sp was measured at Alert by using a TSI nephelometer at three different wavelengths; 450, 550 and 700 nm. The truncation error of the nephelometer, which is due to an angular integration restriction to 7 and 170o (Anderson and Ogren, 1998) was estimated and applied to scattering measurements. Scattering correction was applied to absorption measurements as shown in equation 4."

Detailed comments: P1, L3 use the same sig figs. Not sure what the reviewer means for P1 L3, you mean P2, L3 where I mentioned 8±4 etc. Yes fixed to two sig. figs. 8.0±4.0, 8.0±4.0, 5.0±2.5 and 9.0±4.5. You mention P3, line 9 but it is P3, L19, changed to "...light attenuation is converted to BC using a MAC value..." P6, L15 ECCC? It is an acronym for Environment and Climate Change Canada, name of our government organization. P9, L17, you mean P9 L25 dATN cannot be >2 if you use the definition of ATN in EQ. (1) – the manufacturer presents ATN in percents. "...estimate that for $\Delta$ATN >= 2 a relative uncertainty (dsigma_atn/sigma_atn) is 2.5% (for noise

only)." The $\Delta$ATN>=2 is in "units" as reported by the instrument which are the ATN measurements multiplied by 100 for numerical convenience. Added to text for clarity, P9, L25, "...2%"

P10, L9 " Equation 7 is rewritten in Eq. 6" Should this be "Equation 4 is rewritten in Eq.7"? P10, L17, changed to Equation 4 is rewritten in Eq. 7.

P1, L20, Eq. 9. I don't understand how you get the numbers 0.02, 1.44 and 0.24. The reviewer means P10, line27 ãĂŰ $\Delta\sigma$ãĂŮ_(ap,cal)=$\sigma$_(ap,meas)/(0.02*$\omega$_0/(1-$\omega$_0 )+1.44)ˆ2 *((0.02*$\omega$_0/(1-$\omega$_0 ))ˆ2+(0.24)ˆ2 )ˆ(1/2)

"The constants in equation 4 were derived by Bond et al. (1999) as K1=0.02$\pm$0.02 and K2=1.22$\pm$0.20". These constants were determined in Bond et al., 1999 for the PSAP instrument. I have added this statement to the paper on P11, L1-4,

"Where K1=0.02$\pm$0.02 and K2=1.22$\pm$0.20 and uncertainties in the calibration constants are $\Delta$K1=0.02 (Bond et al., 1999) and $\Delta$K2=0.24 (from Bond et al., 1999 with Ogren 2010 adjustment)."

P10, L25 now on P11, L9, modified to, "... Weekly zeroes are performed on the PSAP at Alert flushing particle-free air for the time period of one to two hours through instrument."

P11, L1-17. The uncertainty analysis of the SP2 is very short compared with that if the PSAP. The numbers are simply given. Can you give any more details? Is, for instance the number 19% the uncertainty of the slope? Thank you for pointing this out. P11 L13-31 to P12 L1-4 We are replacing one paragraph "There are a number.....underestimated" with "Uncertainty in the rBC mass derived from SP2 measurements arise from several sources. As described in section 2.2 mass calibration for all three SP2s was carried out using Aquadag as an external standard. Uncertainties in the slopes of the Aquadag calibration curves give rise to uncertainty the rBC mass calculated for each detected particle. This uncertainty is dependent on the individual particle size

and ranges from around 5% for the largest particles to around 35% for the smallest particles (based on the calibration with the largest uncertainty). When the individual particle masses are combined to give a 1-hour mass concentration, the overall mass uncertainty arising from the calibration curves is on average 12%, 11%, and 16%, for SP2#58, #44, and #17 respectively. Another uncertainty in the rBC mass arises from using Aquadag as a standard. Due to the SP2's enhanced sensitivity to Aquadag (discussed in section 2.2), the calibration curves were scaled by dividing by a factor of $0.70 \pm 0.05$. After this correction is applied, the combined 1-hour mass concentration uncertainty (arising from uncertainty in the fits of the calibration curves and from the uncertainty in the Aquadag correction) is 19%, 18%, and 23% for SP2#58, #44, and #17 respectively. Additionally, there is an uncertainty of approximately 12%, 7%, and 15%, for SP2#58, #44, and #17 respectively, which arises from the process of fitting the mass distribution and using this fit to estimate how much rBC mass lies outside the instrument detection range. This results in overall mass uncertainties in the range of 25-38% depending on the instrument used. In some cases calibrations were not carried out over the full detection range of the instrument and had to be extrapolated to higher rBC masses. Uncertainty from this extrapolation is not accounted for; however, the linear correlations between rBC mass and peak height are relatively strong (as shown in Fig. 2) suggesting that this is not a large source of error."

P13, L13-14. "... a factor of 3 higher for all data..." Why is this ratio so different from the slopes _1-2 presented in Table 2? A factor of 3 is arising by using comparisons between EBC/EC and rBC measurements by taking the ratios of all data above detection limit and doing pairwise comparison. Table 2 includes all data in the regression including the below detection limit values as these are regressions. The caption also states that seasonally averaged values include negative values too. In Table 3 if you compare, all data concentration in column 2, where also negative values are included, the ratio EBC/rBC=2.2 and EC/rBC=1.8. These are more comparable to Table 2 regressions. Adding this line to clarify the two in Table 3 caption: "Table 3: Statistical parameters such as mean, median and standard deviation for all data, Mar 2011 to

Dec 2013 and data with only above detection limit values included. The ratios are only meaningful for data above the detection limit values. Also pairwise statistics available for data set when both pairs in comparison had data. In comparison to Table 2 where all data including negative values are used in the regression analyses, Table 3 values in first column labelled "All data conc." includes negative values too and are more comparable. Also added in Summary and Conclusions, P23 L6, "....on average results of pairwise analyses show both...."

P15, L8, L10. What is "rate of thickness increase" and "thickness rate"? Replacing on P15 L23-26"As rBC cores decreases below 115 nm, thicker coatings are required to produce a measurable scattering signal. In all panels, the apparent coating thickness increases with decreasing rBC core. Below about 120 nm, the apparent rate of thickness increase with decreasing rBC is larger than above 120 nm. Since only thicker coatings can be measured at smaller sizes, some of the increase in thickness rate is attributed to bias in the elastic scattering detection system toward thicker coatings when the rBC cores are less than 115 nm. Overall, there is some increase in coating thickness with smaller rBC core sizes, but the thicknesses represented at rBC core sizes of less than 120 nm are overestimated." With "In all panels, the apparent coating thickness increases with decreasing rBC core. As rBC cores decrease below 115 nm, thicker coatings are required to produce a measurable scattering signal. As a result, when the rBC cores are less than 115 nm, the coating thickness is overestimated due to bias in the elastic scattering detection system toward thicker coatings." P16, L6. What do you mean by "events"? Event is short for particle detection of different types e.g. incandescent and scattering. P16, L12-L17. Also somewhere earlier: could you estimate the mass or volume fraction of BC in the size range where you do have the coating data?

On P17, L1-2 Changed this statement by adding:

"For rBC cores in the 160-180 nm range, the average particle coating thicknesses in April 2012 and in 15 October-November 2013 were estimated to range from 1.25 to 1.4

(this corresponds to a mass fraction of rBC ranging from 0.51-0.36, assuming a 170 nm rBC core)."

P17, Section 4.1.1 EBC. Here you should give some kind of an estimate of uncertainty due to scattering. See my general comments. Made these changes to section 4.1.1 as per earlier main comments. "EBC will overestimate BC if there is absorption from coexisting components and/or coatings of the rBC cores, such as brown carbon (e.g. Kirchstetter et al., 2004; Lack et al., 2013; Lack and Langridge, 2013) and fine particle dust (Weingartner et al., 2003; Müller et al., 2011). However, the influence of brown carbon may be minimal at Alert as values of the Angstrom Absorption Exponent (AAE) are between 0.5 and 1.5 suggesting predominantly fossil fuel influence (see Supplemental section and Figure_Supplemental_1b). In addition, the Aethalometer response depends on filter loading and multiple scattering by the filter medium and sampled aerosol particles. Scattering correction thus becomes important in cases when the aerosol has higher scattering with respect to total extinction (absorption+scattering), i.e. absorption is low. This is not the case at Alert especially during the Arctic haze time. Summertime measurements could fall into this scenario. Recently, Backman et al. (2017) proposed a reduction of a factor of 3.2 in the light absorption coefficients derived from the Aethalometer due to multiple scattering enhancements associated with particles collected on the filter. These enhancements are considered, at least in part, in the EBC estimate by the ïÅąap value used with the Aethalometer, but there remains uncertainty in ïÅąap, including the use of a constant value for all conditions. EBC (unmodified) needs to be evaluated due to these reasons in comparison to absolute measurements mass techniques."

P17-18, Section 4.1.3 rBC You fit a single lognormal mode. Fine, I would also. However, do you have any info on whether there might be larger, coated BC particles? In all the work we have done with the SP2, I have not seen any data that suggests a larger mode of coated BC. P21, L7. MAAP measures at 637 nm, see Müller et al. See Massling et al., 2015 paper, they state that their aerosol light absorption coefficient was measured

at wavelength of 670 nm. Perhaps they made a mistake in their paper. Changing it to 637 nm in our paper on Pg 22, L9.

Table 2. In some columns there is lin.reg. Are the associated numbers slopes? Why don't you give the slope and offsets and their uncertainties – regression codes give them. Replaced in Table 2, Lin. Reg. with Lin. Reg. $\pm$ std. err. and added slope and std error in the slope to the regressions. Also added in the text on P13, L25, "…intercepts and standard errors,….". Fig 6. The y axes start from < 0 which implies negative thickness. Can't be true. Made these changes to the caption text: "…….Negative coating thickness indicates that the measured scattering for a particle is less than the Mie calculations predict for an rBC particle of that diameter with zero coating. These values arise from the inherent noise in the data, as well as assumptions made about the rBC and coating refractive indices, and particle morphology. "

Added this to the references. Anderson, T.L. and Ogren, J.A.: Determining aerosol radiative properties using TSI 3563 integrating nephelometer, Aerosol Sci. Tech., 29(1), 57-69, 1998.

Please also note the supplement to this comment:
https://www.atmos-chem-phys-discuss.net/acp-2017-339/acp-2017-339-AC1-supplement.pdf

**Supplement:**

**Supplemental section for paper, An Evaluation of three methods for measuring black carbon at Alert, Canada by Sharma et al.**

**Calculation of Aerosol Angstrom Absorption Exponent:**

The aerosol Ångström absorption exponent (AAE) was calculated from the PSAP absorption measurements. The AAE is defined as

$$AAE = \frac{\ln(\frac{\sigma_{ap}(\lambda_1)}{\sigma_{ap}(\lambda_2)})}{\ln(\frac{\lambda_1}{\lambda_2})} \qquad\qquad 1$$

where $\lambda_1$=467 nm and $\lambda_2$=660 nm and $\sigma_{ap}(\lambda_1)$ is absorption at 467 nm and $\sigma_{ap}(\lambda_2)$ is absorption at 660 nm.

**Uncertainty in AAE**

Standard techniques were applied to determine combined uncertainties in the Aerosol Absorption Exponent calculated at two wavelengths; $\lambda_1$=467 nm and $\lambda_2$=660 nm. The uncertainty in AAE is determined by Eq. 2 has also been used in Sherman et al. (2015).

$$\Delta AAE\left(\frac{467nm}{660nm}\right) = ((\frac{\partial AAE}{\partial \sigma_{ap,467}})^2 \Delta\sigma_{ap,467}^2 + (\frac{\partial AAE}{\partial \sigma_{ap,660}})^2 \Delta\sigma_{ap,660}^2 + 2*corr(\sigma_{ap,467}, \sigma_{ap,660})$$

$$* \left(\frac{\partial AAE}{\partial \sigma_{ap,467}}\right) * \left(\frac{\partial AAE}{\partial \sigma_{ap,660}}\right) * \Delta\sigma_{ap,467} * \Delta\sigma_{ap,660}))^{1/2} \qquad (2)$$

where

$$\left(\frac{\partial AAE}{\partial \sigma_{ap,467}}\right) = \frac{2.26}{\sigma_{ap,467}} \quad and \quad \left(\frac{\partial AAE}{\partial \sigma_{ap,660}}\right) = \frac{-2.26}{\sigma_{ap,660}}$$

The time series of hourly light absorption measurements from the PSAP at Alert at 550 nm wavelength is shown in Fig_Supplemental_1a. The light absorption has been corrected according to Bond et al. (1999) and also Ogren (2010) for loading and scattering interferences. Episodic increases in absorption during winter/spring reach as high as 4 Mm$^{-1}$ and overall lower values are measured during the summer and fall. Dust and brown carbon each have strong wavelength dependences, but BC does not. The impact of non-BC light absorbing species will

appear as deviations from near unity (1.1±0.3) in the Ångstrom Absorption exponent (AAE) if the non-BC light absorbing species make up more than 40% of the BC (Lack and Langridge, 2013). At Alert, non-BC light absorbing species may include brown carbon and dust. At Alert, absorbing OC (POC, i.e. brown carbon) is more than 40% of the total absorbing carbon for most of the time.  The hourly averaged AAE values between March 2011 and December 2013 are shown in Fig._Supplemental_1b. Values of AAE between 0.5 and 1.5 represent absorption primarily due to fossil fuel BC. A value near 1.0 is considered to be an example of graphitic carbon particles (Petzold et al., 2009), values between 1 and 1.5 are due to total carbon, while AAE values close to 0.5 may reflect different absorption characteristics of pure elemental carbon and increase with varying amounts of OC (Bahadur et al., 2012). There are brief episodic increases in AAE where values over two are reached, indicating the presence of non-BC absorbing aerosol, but most of the fine mode absorption measurements fall within 0.5-1.5, suggesting that EBC is the primary absorbing component with episodic influences of non-BC absorbing components. Mineral dust gives AAE values of three and larger at visible wavelengths (Petzold et al., 2009), which are not evident in Fig._Supplemental_1b.

[Figure]

[Figure]

**Supplemental_Fig1:** (a) Hourly aerosol light absorption measurements, $\sigma_{ap}$, at 550 nm at Alert; (b) hourly averaged Aerosol Angstrom Exponent (AAE) with uncertainty (light gray) calculated from AAE=-ln($\sigma_{ap}(\lambda_1)/\sigma_{ap}(\lambda_2)$)/ln($\lambda_1/\lambda_2$) where $\lambda_1$=467 nm and $\lambda_2$=660 nm and $\sigma_{ap}(\lambda_1)$ is absorption at 467 nm and $\sigma_{ap}(\lambda_2)$ is absorption at 660 nm. A value of AAE=1 is for graphite aerosol.

[Figure]

**Supplemental_Fig-2:** Improved agreements were obtained between the best estimated black carbon mass and masses obtained by optical technique such as Aethalometer (green and red triangles are for data during spring and winter). EBC Aethalometer and rBC data were averaged to EC sampling times.

---

## Author Comment (AC2) · 7 Sep 2017

Referee D. Baumgardner on "An Evaluation of three methods for measuring black carbon at Alert, Canada" by Sangeeta Sharma et al. First of all, the authors like to thank Dr. D. Baumgardner for accepting to review this paper with such constructive remarks.

The suggestion of including the aerosol Absorption Angstrom Exponent (AAE) is great as we initially included AAE in the earlier version of the paper but it didn't show any distinction between various combustion source influences at Alert location and was thus removed. The hourly average AAE between March 2011 and December 2013 are shown in Figure below. Values of AAE between 0.5 and 1.5 represent absorption primarily due to fossil fuel BC. A value near 1.0 is considered to be an example of

graphitic carbon particles (Petzold et al., 2009), while AAE values close to 0.5 may reflect different absorption characteristics of elemental carbon (Bahadur et al., 2012). There are brief episodic increases in AAE where values over two are reached, indicating the presence of non-BC absorbing aerosol, but most of the fine mode absorption measurements fall within 0.5-1.5, suggesting that BC is the primary absorbing component with episodic influences of non-BC absorbing components. Mineral dust gives AAE values of three and larger at visible wavelengths (Petzold et al., 2009), which are not evident in the Figure.

Bahadur, E., Praveen, P. S., Xu, Y., and Ramanathan, V.: Solar absorption by elemental and brown carbon determined from spectral observations, P. Natl. A. Sci., 109, 17366–17371, 2012. Petzold, A., Rasp, K., Weinzierl, B., Esselborn, M., Hamburger, T., DÌĹornbrack, A., Kandler, K., Schutz, L., Knippertz, P., Fiebig, M., and Virkkula, A.: Saharan dust absorption and refractive index and from aircraft-based observations during SAMUM 2006, Tellus B, 61B, 118–130, 2009.

Since the AAE is clearly and dominantly influenced by fossil fuel combustion, we are adding this Figure_Supplemental_2a showing 550 nm absorption and Figure_Supplemental_2b showing AAE time-series. The method, uncertainty calculation and discussion of AAE has been added to the Supplement and few sentences to support fossil fuel influence on the aerosol in the paper.

P 16, lines 10-13.... The aerosol Absorption Ångstrom Exponent (AAE) values, as discussed in the supplemental section (see Figure_supplemental_1b), also suggests predominately fossil fuel sources of rBC and little biomass burning influence (AAEavg (April and Oct)=0.75±0.12). P 17, lines 18-20... The influence of brown carbon may be minimal at Alert as values of the aerosol Absorption Ångstrom Exponent (AAE) are between 0.5 and 1.5 suggesting predominantly fossil fuel influence (see Figure_Supplemental_1b). P 20, lines 6-8, ... "As discussed earlier, the influence of brown carbon due to biomass burning is minimal at Alert during the Arctic haze time for the data collected during the 2011-2013 (AAEavg for April =

0.75±0.12). Thus, that effect of brown carbon will be minimal on the MAC. Also added this to supplemental section: Calculation of Aerosol Angstrom Absorption Exponent: The aerosol Ångström absorption exponent (AAE) was calculated from the PSAP absorption measurements. The AAE is defined as AAE=(lnâĄą(($\sigma$_ap ($\lambda$_1))/($\sigma$_ap ($\lambda$_2)))/(lnâĄą(âĄąãĂŰ(($\lambda$_1)/$\lambda$_2 ãĂŮ )) 1 where ïĄň1=467 nm and ïĄň2=660 nm and $\sigma$ap(ïĄň1) is absorption at 467 nm and $\sigma$ap(ïĄň2) is absorption at 660 nm.

Uncertainty in AAE Standard techniques were applied to determine combined uncertainties in the Aerosol Absorption Exponent calculated at two wavelengths; ïĄň1=467 nm and ïĄň2=660 nm. The uncertainty in AAE is determined by Eq. 2 has also been used in Sherman et al. (2015).

$\triangle$AAE(467nm/660nm)=(($\partial$AAE/ãĂŰ$\partial\sigma$ãĂŮ_(ap,467)     )ˆ2
$\triangle$ãĂŰ$\sigma$_(ap,467)ãĂŮˆ2+($\partial$AAE/ãĂŰ$\partial\sigma$ãĂŮ_(ap,660)     )ˆ2
$\triangle$ãĂŰ$\sigma$_(ap,660)ãĂŮˆ2+2*corr($\sigma$_(ap,467),$\sigma$_(ap,660)     )
ãĂŰ*($\partial$AAE/ãĂŰ$\partial\sigma$ãĂŮ_(ap,467)     )*($\partial$AAE/ãĂŰ$\partial\sigma$ãĂŮ_(ap,660)
)*$\triangle\sigma$_(ap,467)*$\triangle\sigma$_(ap,660)))ãĂŮˆ(1/2)   (2)   where   ($\partial$AAE/ãĂŰ$\partial\sigma$ãĂŮ_(ap,467)
)=2.26/$\sigma$_(ap,467) and ($\partial$AAE/ãĂŰ$\partial\sigma$ãĂŮ_(ap,660) )=(-2.26)/$\sigma$_(ap,660)

The time series of hourly light absorption measurements from the PSAP at Alert at 550 nm wavelength is shown in Fig_Supplemental_1a. The light absorption has been corrected according to Bond et al. (1999) and also Ogren (2010) for loading and scattering interferences. Episodic increases in absorption during winter/spring reach as high as 4 Mm-1 and overall lower values are measured during the summer and fall. Dust and brown carbon each have strong wavelength dependences, but BC does not. The impact of non-BC light absorbing species will appear as deviations from near unity (1.1±0.3) in the Absorption Ångstrom exponent (AAE) if the non-BC light absorbing species make up more than 40% of the BC (Lack and Langridge, 2013). At Alert, non-BC light absorbing species may include brown carbon and dust. At Alert, absorbing OC (POC, i.e. brown carbon) is more than 40% of the total absorbing carbon for most of the time. The hourly averaged AAE values between March 2011 and December

2013 are shown in Fig._Supplemental_1b. Values of AAE between 0.5 and 1.5 represent absorption primarily due to fossil fuel BC. A value near 1.0 is considered to be an example of graphitic carbon particles (Petzold et al., 2009), values between 1 and 1.5 are due to total carbon, while AAE values close to 0.5 may reflect different absorption characteristics of pure elemental carbon and increase with varying amounts of OC (Bahadur et al., 2012). There are brief episodic increases in AAE where values over two are reached, indicating the presence of non-BC absorbing aerosol, but most of the fine mode absorption measurements fall within 0.5-1.5, suggesting that EBC is the primary absorbing component with episodic influences of non-BC absorbing components. Mineral dust gives AAE values of three and larger at visible wavelengths (Petzold et al., 2009), which are not evident in Fig._Supplemental_1b. Addition of Supplemental_Fig1:

Additional comments: P6, line 7: "There are no other scattering or absorption corrections,..." I don't understand why corrections are not being applied when further on PSAP is corrected. Our main purpose for using the Aethalometer data is in its "unmodified form" to see how well it compares to other more absolute mass techniques for measurement of "black carbon". There is enhancement in the absorption in Aethalometer due to filter matrix as well as scattering components on the aerosol and it could be as much as by a factor of 3 as recently estimated by Backman et al., 2017 for several Arctic locations including Alert (which includes loading and scattering correction). Magee uses a much higher MAC value than needed for aerosol in the atmosphere. A MAC value of 16.6 m2g-1 at 880 nm has been used in the Aethalometer firmware by the manufacturer to compensate for these artifacts and give best estimate of EBC. We wanted to demonstrate how well is Aethalometer measuring EBC with Magee's MAC value used at 880 nm. We have long term trends in the EBC measurements derived from Aethalometer at Alert. In the past, we have compared EBC to EC to apply a correction to EBC at Alert and the two techniques agree quite well during the Arctic haze time. Modified by adding these lines on P6, lines 8-11: "There are no other scattering or loading corrections applied to Aethalometer data because a comparison of unmodified EBC mass to best estimate of "BC" mass values are also determined

in this paper. The enhancement in the absorption due to total scattering has been compensated by using a higher $\alpha$ap value used by the Aethalometer firmware. Also added to Section 4.1.1, pg 17 lines 21-25: "In addition, the Aethalometer response depends on filter loading and multiple scattering by the filter medium and sampled aerosol particles. Scattering correction thus becomes important in cases when the aerosol has higher scattering with respect to total extinction (absorption+scattering), i.e. absorption is low. This is not the case at Alert especially during the Arctic haze time. Summertime measurements could fall into this scenario."

Also Pg 18, lines 1-2: EBC (unmodified) needs to be evaluated due to these reasons in comparison to absolute measurements mass techniques. PSAP absorption coefficients have been corrected by using scattering data measured by a 3-w TSI Nephelometer. The main purpose for applying scattering corrections for PSAP absorption was to derive a MAC values at this location by using best estimate of averaged mass of EC and rBC measurements. The scattering correction was absolutely necessary for this purpose.

Added on P6, lines 30, P7 lines1 &2 Aerosol light scattering, ïĄşsp was measured at Alert by using a TSI nephelometer at three different wavelengths: 450, 550 and 700 nm. The truncation error of the nephelometer, which is due to an angular integration restriction to 7 and 170o (Anderson and Ogren, 1998) was estimated and applied to scattering measurements. Scattering correction was applied to absorption measurements as shown in equation 4.

P7, Line 6: How was PSAP measurements converted from 530 to 550 nm? Added on P7, line6, "by using (ïĄň)-1 relationship to the wavelength. . .."

Section 2.4: The uncertainty estimates should be added in Table 1. Uncertainties were added to column #1.

Section 4.1.4: Should explain why EBC is not used in the best estimate of BC

We haven't included EBC in the best estimate of BC as it is light attenuated inferred mass measurement. Our comparison at the end of the paper tells us that these measurements are very close to best estimated absolute mass measurements. Added to P19, lines 17-18, "Considering all arguments, including EC and rBC being more specific direct mass measurements than EBC, which is light attenuation inferred mass indirect measurement,….."

Added to P19 lines 23-25, "EBC mass is not used in the determination of best estimate mass of "BC" as it is an inferred mass derived from optical measurements and need to be evaluated with more direct mass measurements techniques at Alert, presented in the later section."

Page 21 Supplement figure. Fixed the caption as shown below.

"....(green and red triangles are for data during spring and winter)"

Please also note the supplement to this comment:
https://www.atmos-chem-phys-discuss.net/acp-2017-339/acp-2017-339-AC2-supplement.pdf
* * *
[Figure]

**Supplemental_Fig-2:** Improved agreements were obtained between the best estimated black carbon mass and masses obtained by optical technique such as Aethalometer (green and red triangles are for data during spring and winter). EBC Aethalometer and rBC data were averaged to EC sampling times.

**Fig. 1.**

**Supplement:**

**Supplemental section for paper, An Evaluation of three methods for measuring black carbon at Alert, Canada by Sharma et al.**

**Calculation of Aerosol Angstrom Absorption Exponent:**

The aerosol Ångström absorption exponent (AAE) was calculated from the PSAP absorption measurements. The AAE is defined as

$$AAE = \frac{\ln(\frac{\sigma_{ap}(\lambda_1)}{\sigma_{ap}(\lambda_2)})}{\ln(\frac{\lambda_1}{\lambda_2})} \qquad\qquad 1$$

where $\lambda_1$=467 nm and $\lambda_2$=660 nm and $\sigma_{ap}(\lambda_1)$ is absorption at 467 nm and $\sigma_{ap}(\lambda_2)$ is absorption at 660 nm.

**Uncertainty in AAE**

Standard techniques were applied to determine combined uncertainties in the Aerosol Absorption Exponent calculated at two wavelengths; $\lambda_1$=467 nm and $\lambda_2$=660 nm. The uncertainty in AAE is determined by Eq. 2 has also been used in Sherman et al. (2015).

$$\Delta AAE\left(\frac{467nm}{660nm}\right) = ((\frac{\partial AAE}{\partial \sigma_{ap,467}})^2 \Delta\sigma_{ap,467}^2 + (\frac{\partial AAE}{\partial \sigma_{ap,660}})^2 \Delta\sigma_{ap,660}^2 + 2*corr(\sigma_{ap,467}, \sigma_{ap,660})$$

$$* \left(\frac{\partial AAE}{\partial \sigma_{ap,467}}\right) * \left(\frac{\partial AAE}{\partial \sigma_{ap,660}}\right) * \Delta\sigma_{ap,467} * \Delta\sigma_{ap,660}))^{1/2} \qquad (2)$$

where

$$\left(\frac{\partial AAE}{\partial \sigma_{ap,467}}\right) = \frac{2.26}{\sigma_{ap,467}} \quad and \quad \left(\frac{\partial AAE}{\partial \sigma_{ap,660}}\right) = \frac{-2.26}{\sigma_{ap,660}}$$

The time series of hourly light absorption measurements from the PSAP at Alert at 550 nm wavelength is shown in Fig_Supplemental_1a. The light absorption has been corrected according to Bond et al. (1999) and also Ogren (2010) for loading and scattering interferences. Episodic increases in absorption during winter/spring reach as high as 4 Mm$^{-1}$ and overall lower values are measured during the summer and fall. Dust and brown carbon each have strong wavelength dependences, but BC does not. The impact of non-BC light absorbing species will

appear as deviations from near unity (1.1±0.3) in the Ångstrom Absorption exponent (AAE) if the non-BC light absorbing species make up more than 40% of the BC (Lack and Langridge, 2013). At Alert, non-BC light absorbing species may include brown carbon and dust. At Alert, absorbing OC (POC, i.e. brown carbon) is more than 40% of the total absorbing carbon for most of the time.  The hourly averaged AAE values between March 2011 and December 2013 are shown in Fig._Supplemental_1b. Values of AAE between 0.5 and 1.5 represent absorption primarily due to fossil fuel BC. A value near 1.0 is considered to be an example of graphitic carbon particles (Petzold et al., 2009), values between 1 and 1.5 are due to total carbon, while AAE values close to 0.5 may reflect different absorption characteristics of pure elemental carbon and increase with varying amounts of OC (Bahadur et al., 2012). There are brief episodic increases in AAE where values over two are reached, indicating the presence of non-BC absorbing aerosol, but most of the fine mode absorption measurements fall within 0.5-1.5, suggesting that EBC is the primary absorbing component with episodic influences of non-BC absorbing components. Mineral dust gives AAE values of three and larger at visible wavelengths (Petzold et al., 2009), which are not evident in Fig._Supplemental_1b.

[Figure]

[Figure]

**Supplemental_Fig1:** (a) Hourly aerosol light absorption measurements, $\sigma_{ap}$, at 550 nm at Alert; (b) hourly averaged Aerosol Angstrom Exponent (AAE) with uncertainty (light gray) calculated from AAE=-ln($\sigma_{ap}(\lambda_1)/\sigma_{ap}(\lambda_2)$)/ln($\lambda_1/\lambda_2$) where $\lambda_1$=467 nm and $\lambda_2$=660 nm and $\sigma_{ap}(\lambda_1)$ is absorption at 467 nm and $\sigma_{ap}(\lambda_2)$ is absorption at 660 nm. A value of AAE=1 is for graphite aerosol.

[Figure]

**Supplemental_Fig-2:** Improved agreements were obtained between the best estimated black carbon mass and masses obtained by optical technique such as Aethalometer (green and red triangles are for data during spring and winter). EBC Aethalometer and rBC data were averaged to EC sampling times.

---

## Author Response (AR2)

**Co-Editor Decision: Reconsider after minor revisions (Editor review)** (28 Sep 2017) by Willy Maenhaut
Comments to the Author:
The authors have reasonably addressed the comments of the two anonymous referees and they have modified their manuscript accordingly. However, the comments given below should be addressed and numerous alterations are needed for the Main text and Supplemental section before the manuscript can be published in ACP.

Dear Co-Editor:

We made all the corrections as suggested by you and these are highlighted in yellow below. The changes are made in red in the manuscript. Thank you for taking the time to improve the quality of this paper.

Regards,

Sangeeta.

Main text:
Page 1, line 14, and further throughout the manuscript: Replace "e.g. " by "e.g., ".
Page 1, line 24: Replace "is difficult" by "are difficult".
Page 2, line 3: Replace "of BC" by "of the BC".
Page 3, line 28: Replace "in three" by "at three".
Page 4, line 12: Replace "of instruments" by "of the instruments".
Page 4, line 25: Replace "summarizes list" by "provides a list".
Page 5, line 5: Abbreviations and acronyms, here PSAP, should only be defined once within the text. PSAP was already defined on page 3, lines 28-29.
Page 5, line 14: Replace "of Aethalometer" by "of the Aethalometer".
Page 5, line 18: Replace "by single" by "by a single".
Page 6, line 7: Replace "applied to" by "applied to the".
Page 6, line 8: Replace "to best" by "to the best".
Page 6, line 9: Replace "And that how" by "It is further determined how".
Page 6, line 10: Replace "to total" by "to the total".
Page 6, line 16: Replace "1- hour" by "1-hour".
Page 6, line 19: Abbreviations and acronyms, here PSAP, should only be defined once within the text. PSAP was already defined on page 3, lines 28-29.
Page 6, line 21: Replace "operation to the" by "operation as the".
Page 7, line 3: Replace "to scattering measurements. Scattering" by "to the scattering measurements. The scattering".
Page 7, line 4: Replace "to absorption" by "to the absorption".
Page 7, line 6: Replace "by using" by "by using the".
Page 7, line 29: Replace "on calibration" by "on a calibration".
Page 7, line 29: The source or origin of the Aquadag particles should be provided.

Page 8, line 3: Replace "concentrations" by "concentration".

Page 8, line 8: Replace "are shown" by "is shown".

Page 8, line 16: Abbreviations and acronyms, here VED, should be defined (written full-out) when first used within the text. I do realize that it is defined in the caption of Fig. 2, but it should also be defined here.

Page 8, line 24: Replace "1µm" by "1 µm".

Page 9, line 6: Replace "from analysis" by "from the analysis".

Page 10, line 2: Replace "Louisse" by "Liousse".

Page 10, line 14: Replace "in flowmeter" by "in the flowmeter".

Page 10, line 22: Replace "1- ω" by "1 - ω" and replace "from" by "from the".

Page 11, line 2: Replace "and uncertainties" by "and the uncertainties".

Page 11, line 5: Replace "range" by "ranges".

Page 11, lines 5 and 7-8: It is unclear to me why two values are given in each of these cases and what they denote. This should be clarified.

The measured single scattering albedo is in the range 0.95 to 1 at Alert and measured absorption pertaining to SSA are in the range 0.5 and 1 Mm-1 and thus the uncertainty in absorption coefficient lies between 0.099 and 0.11. So replaced "...uncertainty in the absorption coefficient, which has been calculated to be 0.099...." by "...uncertainty in the absorption coefficient, which has been calculated to be between 0.099...."

Page 11, line 8, and further throughout the manuscript: Replace "i.e. " by "i.e., ".

Page 11, line 9: Replace "for time-period" by "for a time period".

Page 11, line 13: Replace "from SP2" by "from the SP2" and replace "arise" by "arises".

Page 11, line 16: Replace "uncertainty the" by "uncertainty in the".

Page 11, line 25: Replace "from uncertainty" by "from the uncertainty".

Page 12, line 11: Replace "47mm" by "47 mm".

Page 12, line 20: Replace "from field" by "from the field" and replace "of sampling" by "of the sampling".

Page 12, line 21: Replace "of total" by "of the total".

Page 12, line 25: Replace "of EC" by "of the EC".

Page 13, line 4: Replace "Time series" by "The time series".

Page 13, line 30: Replace "on differences in these" by "the differences in the".

Page 14, line 1: Replace "approxim. a" by "approximately a".

Page 14, line 30: Replace "2.26-" by "2.26 -".

Page 15, line 1: Replace "1.5-0.0i" by "1.5 - 0.0i".

Page 15, line 18: Replace "particle." by "particles.".

Page 15, line 25: Replace "As rBC" by "As the rBC".

Page 15, line 30: Replace "of rBC" by "of the rBC" and replace "Fig. 6" by "Fig. 7".

Page 16, line 13: The "=" should not be in subscript.

Page 16, line 15: Replace "and for" by "and for a".

Page 17, line 20: Replace "Figure_Supplemental_1b" by "Supplemental_Fig1b.

Page 18, line 2: Replace "measurements mass" by "measurement mass".

Page 18, line 12: Replace "with POC" by "with the POC".

Page 18, line 14: Replace "that eludes" by "that elutes".

Page 18, line 16: Replace "from POC+CC" by "from the POC+CC".

Page 18, line 17: Replace "symbol" by "symbols" twice and replace "data could" by "data (Fig. 8) could".

Page 18, line 19: Replace "symbol" by "symbols".

Page 18, line 23: Replace "by SP2" by "by the SP2".

Page 18, line 24: Replace "relatively to" by "relative to".

Page 19, line 11: Replace "As above" by "As indicated above".

Page 19, line 15: Replace "to their" by "to the".

Page 19, line 28: "MAC" was already defined on page 3, line 5; it should not be defined again. Therefore; replace "The mass absorption coefficient (MAC) is" by "The MAC".

Page 20, line 7: Replace "during the" by "during".

Page 20, line 8: Replace "Uncertainty" by "The uncertainty".

Page 20, line 17: Is "less than 75%" correct here? Should it not be "greater than 75%"?

Yes it should be greater than for exclusion criteria. Changed to "…greater…".

Page 20, line 19: Replace "respectively for" by "respectively, for".

Page 20, line 20: Replace "Evident in" by "As is evident in".

Page 20, line 22: Replace "Second, the" by "Secondly, the".

Page 20, line 23: Replace "reduce relative" by "reduce the relative".

Page 20, line 25: Replace "datum" by "data point".

Page 21, line 3: Abbreviations and acronyms, here OM, should be defined (written full-out) when first used within the text.

Page 21, line 22: Replace "shell-core" by "core-shell" for consistency.

Page 21, line 23: Replace "to presence" by "to the presence".

Page 22, line 2: Replace "over" by "over a".

Page 22, line 6: Replace "Alert, located" by "At Alert, located" and replace "a general" by "the general".

Page 22, line 12: Replace "from MAAP" by "from the MAAP".

Page 22, line 16: It is unclear who or what is meant by "They".

Replaced "They…" by "The European study…"

Page 22, line 18: Replace "Kondo et al., 2011" by "Kondo et al. (2011)".

Page 22, line 21: Replace "and the" by "and".

Page 22, line 27: Replace "for 2012" by "for the 2012".

Page 22, line 28: Replace "EBC from" by "EBC from the".

Page 23, line 25: Replace "to their" by "to the". Now on pg 24 line 2

Page 24, line 12: Replace "uncertainty are" by "uncertainty and the results were".

Page 24, line 13: Replace "values are further" by "values were further".

Page 24, line 14: Replace "and obtained" by "and this resulted in".

Page 24, line 24: Replace "providing the assistance" by "providing assistance".

Page 24, line 26: Replace "Thanks to" by "Thanks are due to".

Pages 25-30, References: Titles of journal articles should all be in lower case; a few are incorrectly in

Title case.

Page 25, line 10: Replace "submitted Atmos." by "submitted to Atmos.".

 The paper has been accepted so changed "…submitted…" by "…accepted in…."

Page 29, line 13: Replace "Tellus" by "Tellus B".

Pages 31-33: The table headings belong above the tables.

Page 31, heading of Table 1: Replace "instrument" by "instruments".

Page 31, Table 1: Within the last column, replace "Mar 2012" by "Mar., 2012" for consistency.

Replaced all abbreviations for months by a dot at the end of the month.

Page 32, line 3: It is unclear to me why the term "difference" is used. Should "interquartile difference reported at 25 and 75 percentile" not simply be replaced by "25 and 75 percentile values"?

Replaced "…difference…" by "…values…"

Page 32, last line: Replace "of values" by "of the values".

Page 33, line 4: Replace "for data" by "for the data".

Page 33, line 5: Replace "in comparison" by "in the comparison".

Page 33, line 6: Replace "in first" by "in the first".

Page 35, caption of Figure 2, line 2: Replace "are response to" by "apply to the".

Page 35, caption of Figure 2, line 3: Replace "and in red circles are SP2 response to" by "and those in red circles to the".

Page 35, caption of Figure 2, line 4: Replace "shows volume" by "shows the volume".

Page 36, caption of Figure 3, line 6: Replace "shows mode" by "shows a mode".

Page 36, caption of Figure 3, line 7: Replace "giving smaller" by "giving a smaller".

Page 36, caption of Figure 3, line 8: Replace "diameter VED" by "diameter. VED" and replace "represents size" by "represent the size".

Page 37, caption of Figure 4, line 1: Replace "A typical" by "Typical".

Page 37, caption of Figure 4, line 4: Replace "however it" by "however, it".

Page 37, caption of Figure 4, line 5: Replace "determine OC" by "determine the OC".

Page 37, caption of Figure 4, line 6: Replace "of OC" by "of the OC".

Page 38, caption of Figure 5, line 2: Replace "from Aethalometer" by "from the Aethalometer".

Page 38, caption of Figure 5, line 3: Replace "mass and of" by "mass of".

Page 39, caption of Figure 6, line 7: Replace "as well as" by "as well as from".

Page 41, caption of Figure 8, line 1: Replace "A weak" by "The weak".

Page 41, caption of Figure 8, line 3: Replace "show some" by "shows some".

Page 41, caption of Figure 8, line 4: Replace "co-eluding thermal" by "co-eluting thermal".

Page 43, caption of Figure 10, line 3: Replace "circle) than November (green star), also" by "circles) than in November (green stars); also".

Page 43, caption of Figure 8, line 5: Replace "Bergstrom, 2006" by "Bergstrom (2006)".

Supplemental section:

There are a number of literature references in this section. A Reference list is therefore needed.

6 References have been added.

Page 1, line 4: Replace "of Aerosol" by "of the Aerosol".

Page 1, first line below equation (1): On two occasions, replace "is absorption" by "is the absorption".

Page 1, fourth line below equation (1): Replace "determine combined" by "determine the combined".

Page 1, sixth line below equation (1): Replace "2 has also" by "2 and has also".

Page 1, line 5 from below: Replace "Fig_Supplemental_1a" by "Supplemental_Fig1a".

Page 2, line 4: Replace "i.e." by "i.e.,".

Page 2, line 6: Replace "Fig._Supplemental_1b" by "Supplemental_Fig1b".

Page 2, last line: Replace "Fig._Supplemental_1b" by "Supplemental_Fig1b".

Page 4, caption of Figure, line 2: Replace "by optical technique such as" by "by an optical technique such as the".

[revised manuscript text omitted]

**Calculation of the Aerosol Angstrom Absorption Exponent:**

The aerosol Ångström absorption exponent (AAE) was calculated from the PSAP absorption measurements.  The AAE is defined as

$$AAE = \frac{\ln(\frac{\sigma_{ap}(\lambda_1)}{\sigma_{ap}(\lambda_2)})}{\ln(\frac{\lambda_1}{\lambda_2})} \qquad (1)$$

where $\lambda_1$=467 nm and $\lambda_2$=660 nm and $\sigma_{ap}(\lambda_1)$ is the absorption at 467 nm and $\sigma_{ap}(\lambda_2)$ is the absorption at 660 nm.

**Uncertainty in AAE**

Standard techniques were applied to determine the combined uncertainties in the Aerosol Absorption Exponent calculated at two wavelengths; $\lambda_1$=467 nm and $\lambda_2$=660 nm. The uncertainty in AAE is determined by Eq. 2 and has also been used in Sherman et al. (2015).

$$\Delta AAE\left(\frac{467nm}{660nm}\right) = ((\frac{\partial AAE}{\partial \sigma_{ap,467}})^2 \Delta\sigma_{ap,467}^2 + (\frac{\partial AAE}{\partial \sigma_{ap,660}})^2 \Delta\sigma_{ap,660}^2 + 2 * corr(\sigma_{ap,467}, \sigma_{ap,660})$$

$$* \left(\frac{\partial AAE}{\partial \sigma_{ap,467}}\right) * \left(\frac{\partial AAE}{\partial \sigma_{ap,660}}\right) * \Delta\sigma_{ap,467} * \Delta\sigma_{ap,660}))^{1/2} \qquad (2)$$

where

$$\left(\frac{\partial AAE}{\partial \sigma_{ap,467}}\right) = \frac{2.26}{\sigma_{ap,467}} \; and \; \left(\frac{\partial AAE}{\partial \sigma_{ap,660}}\right) = \frac{-2.26}{\sigma_{ap,660}}$$

The time series of hourly light absorption measurements from the PSAP at Alert at 550 nm wavelength is shown in Supplemental_Fig1a. The light absorption has been corrected according to Bond et al. (1999) and also Ogren (2010) for loading and scattering interferences. Episodic increases in the absorption during winter/spring reach as high as 4 Mm[-1] and overall lower values are measured during the summer and fall. Dust and brown carbon each have strong wavelength dependences, but BC does not. The impact of non-BC light absorbing species will

appear as deviations from near unity (1.1±0.3) in the Ångstrom Absorption exponent (AAE) if the non-BC light absorbing species make up more than 40% of the BC (Lack and Langridge, 2013). At Alert, non-BC light absorbing species may include brown carbon and dust. At Alert, absorbing OC (POC, i.e., brown carbon) is more than 40% of the total absorbing carbon for most of the time. The hourly averaged AAE values between March 2011 and December 2013 are shown in Supplemental_Fig1b. Values of AAE between 0.5 and 1.5 represent the absorption primarily due to fossil fuel BC. A value near 1.0 is considered to be an example of graphitic carbon particles (Petzold et al., 2009), values between 1 and 1.5 are due to total carbon, while AAE values close to 0.5 may reflect different absorption characteristics of pure elemental carbon and increase with varying amounts of OC (Bahadur et al., 2012). There are brief episodic increases in the AAE where values over two are reached, indicating the presence of non-BC absorbing aerosol, but most of the fine mode absorption measurements fall within 0.5-1.5, suggesting that EBC is the primary absorbing component with episodic influences of non-BC absorbing components. Mineral dust gives AAE values of three and larger at visible wavelengths (Petzold et al., 2009), which are not evident in Supplemental_Fig1b.

[Figure]

[Figure]

**Supplemental_Fig1:** (a) Hourly aerosol light absorption measurements, $\sigma_{ap}$, at 550 nm at Alert; (b) hourly averaged Aerosol Angstrom Exponent (AAE) with uncertainty (light gray) calculated from AAE$=-\ln(\sigma_{ap}(\lambda_1)/\sigma_{ap}(\lambda_2))/\ln(\lambda_1/\lambda_2)$ where $\lambda_1$=467 nm and $\lambda_2$=660 nm and $\sigma_{ap}(\lambda_1)$ is absorption at 467 nm and $\sigma_{ap}(\lambda_2)$ is absorption at 660 nm. A value of AAE=1 is for graphite aerosol.

[Figure]

**Supplemental_Fig2:** Improved agreements were obtained between the best estimated black carbon mass and masses obtained by an optical technique such as Aethalometer (green and red triangles are for data during the spring and winter). EBC Aethalometer and rBC data were averaged to EC sampling times.

---

## Author Response (AR3)

**Co-Editor Decision: Publish subject to minor revisions (review by editor)** (21 Oct 2017) by Willy Maenhaut

Dear Editor,

We have made all the grammatical and other changes as recommended by the co-editor. Pls see my response below.

Best regards,

Sangeeta

**Co-Editor Decision: Publish subject to minor revisions (review by editor)** (21 Oct 2017) by Willy Maenhaut
Comments to the Author:
A number of alterations are still needed for the Main text and Supplemental section before the manuscript can be published in ACP.

Main text:
Page 11, line 9: Replace "for time-period" by "for a time period".
Page 13, line 2: Replace "of EC" by "of the EC".
Page 13, line 7: Replace "3.1 The time series and seasonal variations" by "3.1 Time series and seasonal variation".
Page 14, line 4: Replace "elucidate on the" by "elucidate the".
Page 14, line 5: Replace "i.e." by ", i.e.,".
Page 16, line 4: Replace "of rBC particles" by "of the rBC particles".
Page 17, line 26: Replace ", i.e." by ", i.e.,".
Page 20, line 16: Replace "(i.e." by "(i.e.,".
Page 20, line 20: Replace "greaterthan" by "greater than".
Page 20, line 23: Replace "respectively for" by "respectively, for".
Page 21, lines 26 and 29: Replace "shell-core" by "core-shell" for consistency.
Page 22, line 18: Replace "from MAAP" by "from the MAAP".
Page 22, line 22: Replace "study report" by "study reports".
Page 24, line 21: Replace "inwinter" by "in winter".
Pages 25-31, References: Titles of journal articles should all be in lower case; a few are incorrectly in Title Case.

Name of the programs should be capitalized, is this correct?

Fixed these references:

[revised manuscript text omitted]

Page 29, line 5: Replace "61B" by "61".

Page 43, caption of Figure 10, line 3: Replace "black circle" by "black circles". fixed

Supplemental section:
Why does this section start with page number "45"? If it is meant to continue the page numbering from the Main text, then it should start with page number "44". However, I strongly suggest to start with numbering the pages of this section with "1" or "S1" as the numbering in the Main text may be different in the published ACP version. Also, the last three pages have the same page number "45".

Changed to page number starting at 1 in the supplemental section.

Page 45, reference to "Ogren, 2010": The title of the journal article should be in lower case and not in Title Case. Fixed

Page 45, line 2 of reference to "Ogren, 2010": Delete "1608" at the end of the line. fixed

Page 45, line 3 of reference to "Ogren, 2010": Delete "2010.1609,". Fixed

Page 45, line 3 of reference to "Petzold et al., 2009": Replace "61B" by "61". fixed

Last page, caption of Figure, line 2: Replace "such as" by "such as the". fixed

[revised manuscript text omitted]

---

## Author Response (AR4)

**Co-Editor Decision: Publish subject to minor revisions (review by editor)** (27 Oct 2017) by Willy Maenhaut

Dear Dr. Willy Maenhaut,

I have corrected the suggested changes.

Thx for your valuable time and effort in noticing such detailed grammatical errors!

Regards,

Sangeeta.

Comments to the Author:
A few alterations are still needed for the Main text and Supplemental section before the manuscript can be published in ACP.

Main text:
Page 7, line 20: replace "time-periods" by "time periods". Done
Page11, line 9: replace "timeperiod" by "time period".  Done
Page 13, line 7: Replace "3.1 Time series and seasonal variations" by "3.1 Time series and seasonal variation".  done
Page 23, line 3: replace "time-period" by "time period". Done
Page 29, line 3: The name "Dörnbrack" is misspelled. Done

Supplemental section:
Page 2, first line of reference to "Petzold et al., 2009": The name "Dörnbrack" is misspelled. done

[revised manuscript text omitted]